# Identifying single molecule force spectroscopy data using deep learning with physics augmentation

## Abstract

Deciphering the pathways of protein folding and unfolding under tension is essential for deepening our understanding of fundamental biological mechanisms. Such insights offer the potential to develop treatments for a range of incurable and fatal debilitating conditions, including muscular disorders like Duchenne Muscular Dystrophy and neurodegenerative diseases such as Parkinson's disease. Single molecule force spectroscopy (SMFS) is a powerful technique for investigating forces when domains in proteins fold and unfold. Currently, manual visual inspection remains the primary method for classifying force curves resulting from single proteins; a time-consuming task demanding significant expertise. In this work, we develop a classification strategy to detect measurements arising from single molecules by augmenting deep learning models with the physics of the protein being investigated. We develop a novel physics-based Monte Carlo engine to generate simulated datasets comprising of force curves that originate from a single molecule, multiple molecules, or failed experiments. We show that pre-training deep learning models with the simulated dataset enables high throughput classification of SMFS experimental data with average accuracies of $75.3 \pm 5.3\%$ and ROC-AUC of $0.87 \pm 0.05$. Our physics augmentation strategy does not need expensive expert adjudication of the experimental data where models trained using our strategy show up to $25.9\%$ higher ROC-AUC over the models trained solely on the limited SMFS experimental data. Furthermore, we show that incorporating a small subset of experimental data ($\sim 100$ examples) through transfer learning improves accuracy by $6.8\%$ and ROC-AUC by $0.06$. We have validated our results on three new SMFS experimental datasets. To facilitate further research in this area, we make our datasets available and provide a Python-based toolbox (https://anonymous.4open.science/r/AFM_ML-2B8C).

## 1 Introduction

Many biological processes depend on controlling mechanical forces achieved via the folding and unfolding of domains in molecules like titin (Rief et al., 1997; 1998; Oberhauser et al., 2000), dystrophin and its homologue utrophin (Rajaganapathy et al., 2019; Ramirez et al., 2023), neurotoxic proteins (Hervás et al., 2012), and extracellular matrix protein tenascin (Oberhauser et al., 1998). For example, dystrophin and utrophin work as molecular shock absorbers that limit myofiber membrane damage when undergoing reversible unfolding upon muscle stretching and contraction Ervasti (2007). Evidently, studying mechanical properties of single proteins can provide vital insights for unravelling mechanisms of diseases, that include neurodegenerative disorders (Hervás et al., 2012; Ruggeri et al., 2018) and muscle degeneration disease Duchenne Muscular Dystrophy (Ramirez et al., 2023; Hoffman et al., 1987). Instruments such as optical tweezers (Ashkin et al., 1986) and atomic force microscopes (AFMs) (Binnig et al., 1986) have enabled single molecule force spectroscopy (SMFS), where molecular forces in the femto to nano-Newton range, over sub-nanometer to micrometer distances, can be measured and studied. In an SMFS experiment, measurements from a force probe are recorded while it is made to interact with molecules of interest. Since the size of a typical force probe is orders greater than a typical bio-molecule, the probe may come in contact with one or more molecules. Measurements obtained from interactions involving more than one molecule confound

the interpretation of results and are undesirable in characterizing the behavior of single molecules. Thus, identifying measurements that originate from single molecules is important in SMFS.

The use of chemical functionalization of probes, along with molecular fingerprints has emerged as an approach for identifying single molecule force curves (Yang et al., 2020). Chemical functionalization modifies the surface of the force probe and substrate to enable site-specific attachment of proteins, where fingerprints are well-characterized molecules that yield distinct unfolding patterns. However, surface chemical functionalization is time-consuming, often taking at least 6 hours (Zimmermann et al., 2010), and demands careful handling and practice, as probes are delicate and easy to break. Here it is possible for multiple molecules to attach to the probe; however, fingerprints in the force curves can be leveraged to discern single molecule force curves from force curves that result from multiple molecules.. Moreover, advanced filtering techniques, informed by an understanding of all molecules involved in the complex, are essential for effectively identifying single molecule force curves (Yang et al., 2020).

In contrast, conducting experiments without functionalizing probes and introducing fingerprints into the native protein has significant advantages. Probes without functionalization are easier and less expensive to manufacture, and bio-molecules without fingerprints engineered into their structure are easier to synthesize. Moreover, there is greater confidence that the experimental data characterizes the unaltered native bio-molecule without any confounding effects introduced by fingerprints. Despite these advantages, when there are no fingerprints, distinguishing data that originate from single molecules and multiple molecules is more challenging. Currently, the most widely accepted method for distinguishing the data is based on visual inspection, which is a time-consuming process that demands a high level of expertise (Bornschlögl and Rief, 2011; Lyubchenko, 2018). Additionally, force curves need to be collected from a large number of experiments, not only to ensure statistical confidence but also because the protein concentration is generally lowered to minimize the possibility of multiple molecules (Ramirez et al., 2023; Oberhauser et al., 2000). These factors make it challenging to obtain precise statistics of single molecular force curves and to generate a large, annotated dataset appropriate for training deep learning models.

To address these challenges, we develop an effective classification strategy, where we augment deep learning models with the physics of protein unfolding, that can accurately classify force curves from real physical SMFS experiments into three classes: 1) no molecule, 2) single molecule, and 3) multiple molecules. We propose a physics-based Monte Carlo simulation algorithm to generate a large, well-annotated, and balanced dataset for model training. We also test our model performances on three new SMFS experimental datasets, obtained from non-specific pulling of multi-domain molecules: titin, utrophin, and dystrophin (Hua et al., 2024). Our results show deep learning models pre-trained with simulated datasets achieve average accuracies of $75.3 \pm 5.3\%$ and area under the receiver operating characteristic curve (ROC-AUC) (Fawcett, 2006) of $0.87 \pm 0.05$ across three SMFS experimental datasets. Our physics augmentation strategy not only lessens the need for expensive annotations from experts but also outperforms the models trained from limited SMFS experimental datasets directly, achieving up to $25.9\%$ higher ROC-AUC. Additionally, the accuracy and ROC-AUC can be further improved by $6.8\%$ and $0.06$, respectively, when incorporating a small subset of experimental data ($\sim 100$ examples) via the transfer learning technique (Pan and Yang, 2010). Furthermore, this work presents the first publicly accessible SMFS experimental datasets derived from non-specific pulling of three unique multi-domain proteins and associated codes.

## 2 RELATED WORK

**Single molecule classification** A 1D convolutional neural network trained using a triplet loss function (Hoffer and Ailon, 2018) was utilized to classify single molecule force curves into single, multiple, or no molecule classes, with reported accuracy ranging from $65 - 70\%$ (Waite et al., 2023). Moreover, a machine learning workflow was proposed to iteratively classify different unfolding pathways of single molecule curves, achieving high accuracy after sufficient iterations (Doffini et al., 2023). However, these two datasets were collected with chemically functionalized probes and fingerprints. The chemical functionalization process is dependent on the specific protein and thus cannot be made agnostic to the protein under investigation. Moreover, in the first dataset, each single molecule curve contains only one unfolding event (Waite et al., 2023), simplifying the classification problem; the second dataset comprises images rather than time series data (Doffini et al., 2023),

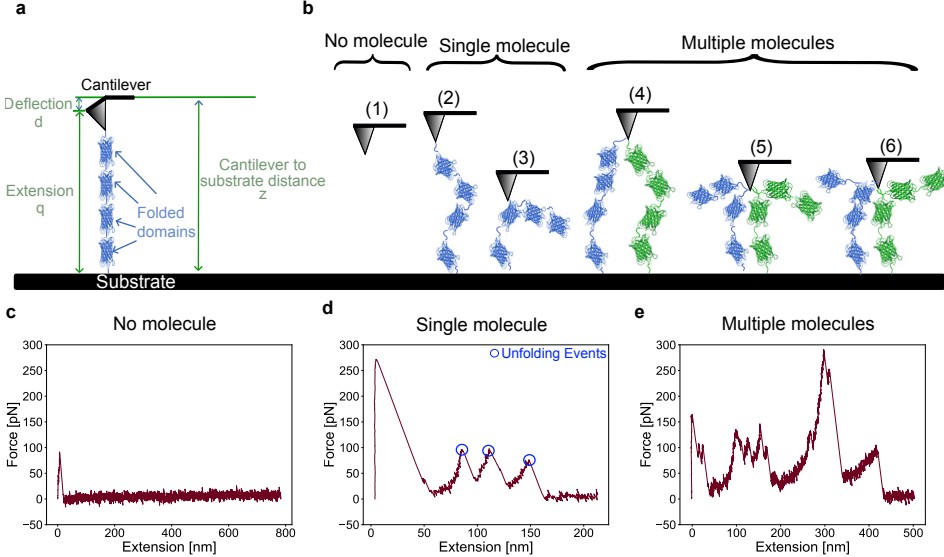

Figure 1: Illustration of AFM based SMFS. (a) Schematic showing the desirable case of a single protein molecule with four folded domains under tension between the tip of the AFM cantilever and the substrate. The deflection $d$ and the separation between the cantilever and the substrate $z$ are measured. The tensile force on the protein is computed from the deflection $d$. (b) Depictions of possible scenarios, categorized into three classes: (1) no molecule present between the tip and substrate, (2-3) a single molecule or a section of a single molecule present between the tip and the substrate, and (4-6) multiple molecules or sections of multiple molecules between the tip and the substrate. (c-e) Show example force curves representative of the three different classes, with blue circles highlighting unfolding events.

yet which introduces redundancy given that force curves are inherently time series data. There are currently no available time series datasets from non-specific pulling of multi-domain proteins. Here, we construct such datasets for classification purposes.

**Time series classification (TSC)** More than hundreds of time series classification (TSC) algorithms, including both non-deep learning methods (Bagnall et al., 2017) and deep learning methods (Fawaz et al., 2019; Wang et al., 2016), are present in prior-art. Although more than 80 different datasets from the University of California, Riverside (UCR) time series classification repositories (Dau et al., 2019) are evaluated with these methods, none of these datasets include SMFS data. Moreover, methods that do not use deep learning become computationally intensive and impractical to execute on large-scale datasets (Bagnall et al., 2017; Fawaz et al., 2019). In this article, we leverage deep learning methods to classify our SMFS datasets.

**Data augmentation** Data augmentation is shown to be successful in addressing the challenges posed by limited data and reducing overfitting in time series datasets (Iwana and Uchida, 2021). Existing data augmentation techniques primarily focus on manipulation of magnitude and timescale information already present in the data, without explicitly incorporating information from the underlying physics that generates the time series. In our study, we propose a strategy that employs the physics of protein unfolding behavior toward data augmentation; we corroborate the effectiveness of such an approach in Section 6.

## 3 SMFS PRELIMINARIES

We first describe an atomic force microscope (AFM) based SMFS setup that is employed to obtain force curves of proteins with multiple domains. Here, a microcantilever with a sharp tip is pressed against a substrate on which the proteins under study are deposited. Under applied force, parts of one or more protein molecules are non-specifically attached to the cantilever tip; characterized by

a stochastic adhesion event (Leite et al., 2012). Upon retraction of the cantilever from the surface, sections of the proteins between the tip of the cantilever and the substrate experience a tensile force. The record of the force experienced by the cantilever (and therefore the protein) versus protein extension $q$ is known as a force curve, as depicted in Figure 1c-e. If only one protein molecule is present between the cantilever tip and substrate, the force curve unveils important mechanical properties of the protein molecule. We illustrate such a scenario in Figure 1a, where a single protein molecule with four folded domains is attached between the substrate and the tip of the cantilever. As the cantilever retracts, the protein experiences mechanical tension, causing the stochastic unfolding of a folded domain. The applied force drops abruptly when a domain unfolds, as highlighted by the blue circles in Figure 1d. This process continues until either all domains are unfolded or the connection between the cantilever and the substrate is broken (which is a stochastic detachment event (Rief et al., 1997)), producing a saw-tooth pattern of force curves (Figure 1d).

In practice, the force curves can be categorized into one of three classes - 1) **No molecule**: where no molecule is present between the tip and substrate, 2) **Single molecule**: when only a single molecule or a section of a single molecule is present, or 3) **Multiple molecules**: where multiple molecules or sections of multiple molecules are present between the tip and substrate. Example experimental force curves corresponding to these three classes are depicted in Figure 1c, 1d, and 1e respectively. The *single molecule* class may include traces with a portion of multi-domain protein (Figure 1b (2,3)); the *multiple molecules* class may involve a combination of different numbers of domains (Figure 1b (4-6)). The *no molecule* traces do not contain useful information about the protein molecule. The force curves originating from multiple molecules typically exhibit larger unfolding forces than those with a single molecule (Figure 1e) and have a mixture of unfolding events that cannot be traced back to a specific protein molecule, confounding useful interpretation. Therefore, excluding force curves with no molecule and multiple molecules is necessary to obtain accurate and interpretable data from SMFS. The identification of the *single molecule* force curves is challenging due to a number of reasons: 1) a large number of force curves (1000-5000) need to be collected in a single experiment since protein capture success rates are kept at 1-5% (Oberhauser et al., 2000; Ramirez et al., 2023), 2) the study of a specific protein involves at least three replications for confidence on results, 3) the force curves are corrupted with substantial noise both from instrument measurement noise and intrinsic thermal noise of the molecules, and 4) often the proteins under investigation have no prior characterization which makes the adjudication time consuming and difficult even for domain experts.

## 4 METHODS

### 4.1 SIMULATING PROTEIN UNFOLDING

Monte Carlo Simulation based methods are used widely in the SMFS studies, yielding results that closely align with experimental data (Liu et al., 2020; King et al., 2010). However, prior simulation frameworks are restricted to the idealized case of force curves arising from single molecule. To build a comprehensive training dataset, we incorporate the real cases of force curves generated by no molecule and multiple molecules. Additionally, we model the cases where only partial sections of molecules are present as well as the stochastic adhesion and detachment events, of the cantilever to the protein, to better approximate experimental data.

For the simulations, $N$ proteins are considered with $i$-th protein having $D^{(i)}$ folded domains attached between the substrate and the force probe. The base of the cantilever probe is moved away at a constant speed $v$; here the position of the base of the cantilever $z$ is initialized at zero and is updated every $\Delta t$ seconds. The protein extension $q$ is determined by solving the equation,

$$k_c d = \sum_{i=1}^{N} F_{WLC}^{(i)}(q, L_c^{(i)}, L_p^{(i)}),$$ (1)

where $k_c$ is the spring constant of the cantilever, and $d$ is the deflection (Fig 1a). Here, $F_{WLC}^{(i)}(q, L_c^{(i)}, L_p^{(i)})$ is the worm-like chain (WLC) model that relates the force applied to the extension of $i$-th protein, given by (Rief et al., 1999),

$$F_{WLC}^{(i)}(q, L_c^{(i)}, L_p^{(i)}) := \frac{k_B T}{L_p^{(i)}} \left[ \frac{1}{4 \left(1 - \frac{q}{L_c^{(i)}}\right)^2} - \frac{1}{4} + \frac{q}{L_c^{(i)}} \right],$$ (2)

---

**Algorithm 1** Monte Carlo Simulation

---

**Require:** $v, L_c^{(i)}, L_p^{(i)}, \Delta L_c^{(i)}, \Delta L_p^{(i)}, k_0, \Delta x^{\ddagger}, \Delta G^{\ddagger}, N, D^{(i)}$

    *Initialization*: $z \leftarrow 0, U^{(i)} \leftarrow 0$

1: **for** $t \leftarrow 0 : \Delta t : T$ **do**
2:    $z \leftarrow z + v\Delta t$
3:    Solve (1) for $q$
4:    **for** $i \leftarrow 1 : 1 : N$ **do**
5:       Calculate $F_{WLC}^{(i)}$ using (2)
6:       Compute $k_{off}(F_{WLC}^{(i)})$ using (5)
7:       Compute $P_u^{(i)}(F_{WLC}^{(i)})$ using (3)
8:       Draw $\eta^{(i)} \sim \mathcal{U}_{[0,1]}$
9:       **if** $\eta^{(i)} < P_u^{(i)}(F_{WLC}^{(i)})$ **then**
10:          $L_c^{(i)} \leftarrow L_c^{(i)} + \Delta L_c, L_p^{(i)} \leftarrow L_p^{(i)} + \Delta L_p$
11:          $U^{(i)} \leftarrow U^{(i)} + 1$
12:       **end if**
13:       Draw $\eta_d^{(i)} \sim \mathcal{U}_{[0,1]}$
14:       **if** $U^{(i)} == D^{(i)}$ and $\eta_d^{(i)} < P_d(F_{WLC}^{(i)})$ **then**
15:          $L_c^{(i)} \leftarrow L_c^{(i)} + C_{Lc}, L_p^{(i)} \leftarrow L_p^{(i)} + C_{Lp}$
16:       **end if**
17:    **end for**
18: **end for**

---

where $k_B$ is the Boltzmann constant, $T$ is temperature, and $L_c^{(i)}$ and $L_p^{(i)}$ are the contour length and the persistence length of the $i$-th protein, respectively. For the $i$-th protein, the probability of a domain unfolding during the time interval $\Delta t$ is found by

$$P_u^{(i)}(F_{WLC}^{(i)}) = (D^{(i)} - U^{(i)})(1 - e^{-k_{off}(F_{WLC}^{(i)})\Delta t}), \tag{3}$$

where $U^{(i)}$ is the number of unfolded domains which is initially set to zero, and $k_{off}(F_{WLC}^{(i)})$ is the transition rate that can be determined with the Dudko-Hummer-Szabo model (Dudko et al., 2008) (See Appendix A.1).

For determining unfolding events of $i$-th protein, a random number $\eta^{(i)}$ is generated uniformly from 0 to 1 and is compared to the unfolding probability $P_u^{(i)}(F_{WLC}^{(i)})$. No unfolding event is triggered if the random number is larger than $P_u^{(i)}(F_{WLC}^{(i)})$; the simulation will continue to the next time slot by adding time interval $\Delta t$. Otherwise, one of the domains is unfolded, leading to a increase in the number of unfolded domains, $U^{(i)}$, by 1, and the simulation continues to the next unfolding event if folded domains still exist after updating contour length and persistence length via adding increments $\Delta L_c^{(i)}, \Delta L_p^{(i)}$ respectively.

Once all domains in $i$-th protein unfold, the protein detaches from either the cantilever tip or substrate based on the detachment probability,

$$P_d(F) = \begin{cases} C_d & F \geq F_{td} \\ 0 & F < F_{td} \end{cases}, \tag{4}$$

where $C_d$ is a constant and $F_{td}$ is a random number sampled from the Gaussian distribution, denoting the threshold at which the detection of detachment begins. Upon detachment of the $i$-th protein, its WLC force is reduced to zero by adding large constants ($C_{Lc}$ and $C_{Lp}$) to contour length $L_c^{(i)}$ and persistence length $L_p^{(i)}$ respectively. To better replicate experimental force curves, we introduce Gaussian noise to the WLC force immediately after its calculation at line 5 of Algorithm 1. The adhesion force (see Appendix A.2) is added at the end of the simulation process.

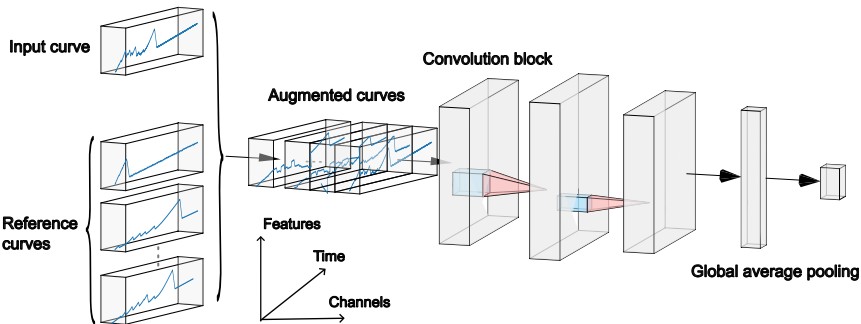

Figure 2: Deep learning models with reference curves, showcasing with Fully Connected Network (FCN) architecture.

### 4.2 AUGMENTING DEEP LEARNING MODELS WITH THE PHYSICS OF PROTEIN UNFOLDING

Given the challenge of constructing a large, well-annotated dataset using SMFS experimental data, we pre-train the deep learning models with simulation data. By utilizing simulation data, we effectively incorporate the underlying physics of protein unfolding into our analysis. Our simulation (Sec 4.1) is carried out using the WLC model. The WLC model encapsulates the physics of the protein unfolding, accurately describing the entropic spring like behavior of the protein between two unfolding events of the protein, which is also corroborated by experimental data. Subsequently, we test the performance of these pre-trained deep learning models using experimental data.

Data augmentation via linear combinations of examples from different classes is shown to be effective in image classification (Summers and Dinneen, 2019; Tokozume et al., 2018a; Huang et al., 2020) and sound recognition (Tokozume et al., 2018b). Here, we incorporate $M \in \mathbb{N}$ reference curves to augment the curve under classification. The reference curves are randomly sampled simulated force curves from the training dataset. Each reference curve $x_i$ is augmented with the input curve $x$ via the difference between two curves $(x - x_i)$ as additional channels, resulting in a total of $M + 1$ channels. The input curve $x$ can be either a simulated or experimental force curve undergoing classification. This augmented input $[x, x - x_1, \cdots, x - x_M]^T$, comprising the input curve and the reference curves, is then passed through deep learning models (Figure 2). The output layer is a softmax layer with three neurons, corresponding to the three classes.

We choose two baseline methods: multi layer perception (MLP) (Wang et al., 2016), which was proposed as a baseline architecture for time series classification (TSC); and the triplet network (Triplet), which was applied on the Discoidin domain receptors (DDRs) dataset (more details in Section 5) (Waite et al., 2023). In addition to these baseline methods, we explore fully connected neural networks (FCN), the residual networks (ResNet), which are two of the highest performing deep neural networks on the UCR time series classification archive (Fawaz et al., 2019). We also investigate the Inception network (InceptionNet) and InceptionTime, an ensemble of five InceptionNet models initialized randomly, which is the current state-of-the-art deep learning model on the UCR archive (Ismail Fawaz et al., 2020). More details of deep learning models can be found in Appendix B.1.

## 5 DATASETS

Here, we construct both simulated and experimental datasets from the non-specific pulling of three multi-domain molecules: Titin I27O, an engineered protein composed of eight repeats of the Ig 27 domain of human titin; utrophin (UtrN-R3) and dystrophin (DysN-R3), fragments encoding the N terminus through spectrin repeat 3. The simulated data were generated using our comprehensive simulation engine, while the experimental data were collected from real physical experiments, as previously described (Hua et al., 2024). For the experimental data, force curves were collected on different days and at various pulling speeds, as outlined in Table 1. As a result, the lengths of the force curves are different, and substantial variations exist even within the same class. Annotation is performed through visual inspection of the unfolding force curves. The comparison between

Table 1: Details of experimental datasets

| Dataset | Number of curves per class | Lengths | Different days | Pulling speeds $[nm/s]$ |
|---|---|---|---|---|
| DDRs (Waite et al., 2023) | [102,102,136] | 400 | NA | 2000 |
| Titin I27O | [181,164,191] | 736-4859 | 5 | 500, 1000, 2000 |
| UtrN-R3 (Hua et al., 2024) | [175,166,200] | 1777-8890 | 11 | 500, 1000, 2000 |
| DysN-R3 (Hua et al., 2024) | [191,185,177] | 1659-4814 | 6 | 500, 1000, 2000 |

simulated and experimental data is discussed in Section B.5. A dataset with $n$ samples $\mathcal{D} = [(\mathcal{F}_1, Y_1), (\mathcal{F}_2, Y_2), \ldots (\mathcal{F}_n, Y_n)]$ is a collection of pairs $(\mathcal{F}_i, Y_i)$, where $\mathcal{F}_i$ is the force data out of the force curve and $Y_i \in \{0, 1, 2\}$ is its label corresponding one of three classes. The force data $\mathcal{F} = [F_1, F_2, \ldots F_T]$ is a sequential set of force values, where $T$ represents the length of $\mathcal{F}$. Additionally, we include the DDRs dataset (Appendix B.2) Waite et al. (2023) to compare the performance of different deep learning methods.

**Titin I27O** Monte Carlo simulations require molecular parameters like contour length, persistence length, and their corresponding increments, which are estimated from experimental data, molecular structure, and energy landscape parameters, which are listed in Table 2. The simulation data, comprising 6400 force curves, is generated for three classes; Class 0 has no protein between the substrate and cantilever tip, Class 1 has one protein, and Class 2 has at least two proteins attached. The number of initially folded domains in each protein $D^{(i)}$ varies from 1 to 8, as illustrated in Figure 6a-c. The experimental dataset is balanced, with approximately 180 force curves for each class, resulting in a total of around 550 force curves (Table 1), with examples illustrated in Figure 6d-f.

**UtrN-R3 and DysN-R3** The simulation datasets for both UtrN-R3 and DysN-R3, each consisting of 6,400 force curves, were generated using parameters in Table 2, and the experimental datasets comprise approximately 180 force curves in each class (Hua et al., 2024). Example curves from both simulation and experiment for UtrN-R3 and DysN-R3 can be found in Figure 7 and Figure 8, respectively. The noise in both UtrN-R3 and DysN-R3 datasets is more prominent compared to Titin I27O dataset as UtrN-R3 and DysN-R3 unfold at smaller forces. Moreover, both UtrN-R3 and DysN-R3 are natural proteins, unlike Titin I27O, which is an engineered protein. These factors make both UtrN-R3 and DysN-R3 datasets more challenging for classification.

## 6 RESULTS AND DISCUSSION

For each of our three protein molecules, Titin I27O, UtrN-R3, and DysN-R3, unless otherwise specified, all deep learning models were pre-trained using 80% of corresponding simulated data for 3-class classification, with the remaining 20% used for validation to monitor overfitting (A detailed discussion of overfitting is provided in C.1). The pre-trained models were subsequently tested on 70% of the experimental datasets, while the remaining 30% was reserved for transfer learning or direct training purposes. Each training was repeated five times, and the model performance was evaluated based on overall accuracy, weighted F1-score, and weighted ROC-AUC. The data underwent preprocessing, including trimming, resampling, and min-max normalization, with additional details provided in Appendix B.3 and implementation details in Appendix B.4. Preprocessing and results for DDRs are provided in Appendix C.2. Unless specified otherwise, one randomly selected reference curve ($M = 1$) was used, as discussed in Section C.3.

We ranked each deep learning model based on its overall accuracy for each dataset, where rank 1 indicates the most accurate model. The datasets consist of both simulated validation sets and experimental testing sets for all protein molecules. The average rank for each model across all datasets was then computed and visualized using a critical difference diagram (Demšar, 2006). In Figure 3, thick horizontal lines represent a group of classifiers that are not significantly different in

terms of accuracy. For statistical analysis, we employed the Wilcoxon signed-rank test with Holm correction as the post-hoc test following the Friedman test (Fawaz et al., 2019; Demšar, 2006).

Our data show that ResNet outperforms the other deep learning models, though its performance is statistically equivalent to to InceptionTime, FCN, and InceptionNet (Figure 3). We selected ResNet for further evaluation, with a focus on its performance on the experimental testing set, as this is our primary objective.

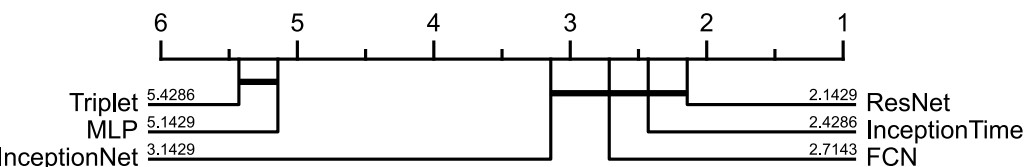

Figure 3: Critical difference diagram of different deep learning models the simulated validation and experimental testing sets for each protein molecule based on average accuracies. The most accurate model is assigned a rank of 1, with a thick horizontal line representing a group of classifiers that do not exhibit statistically significant differences in accuracy.

### 6.1 PHYSICS AUGMENTATION IMPROVES THE PERFORMANCE

We evaluate the physics augmentation strategy, via pre-training with simulation datasets, and compare it to models trained directly from experimental data across three SMFS experimental datasets.

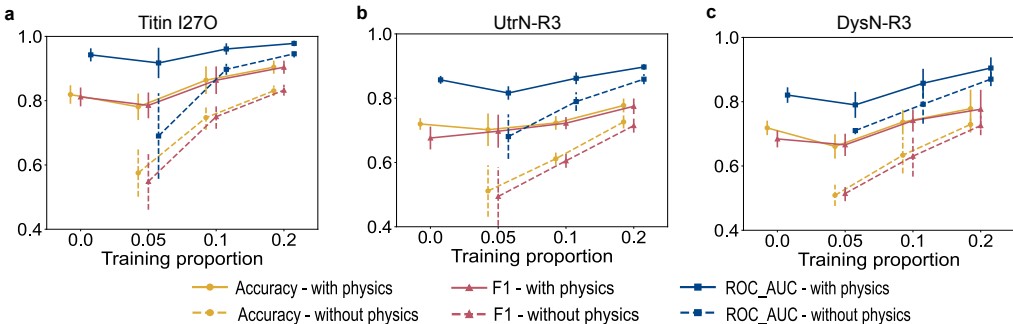

Figure 4: The performance of ResNet trained with different proportions of experimental datasets (training proportion) across three datasets: (a) Titin I27O, (b) UtrN-R3, and (c) DysN-R3. Dashed lines are the results of models trained directly from experimental data, while solid lines represent results achieved via the physics augmentation strategy, with error bars indicating standard deviations over five runs.

We trained deep learning models directly from experimental data, utilizing different proportions ranging from $0.05$ to $0.2$ of the whole dataset. The training data were drawn from $30\%$ of the experimental datasets, while the remaining $70\%$ were reserved as a common testing dataset, with metrics presented as dashed lines in Figure 4. For a training proportion of $0.05$, the models exhibit poor performance, with ROC-AUC of $0.7$ and accuracy of $0.5$ across all three datasets. As the amount of experimental data increases, the performance improves, achieving ROC-AUC of $0.8$ and accuracy of $0.7$ across three datasets.

With the physics augmentation strategy, deep learning models were initially trained with simulation data, and subsequently tested on $70\%$ of the experimental data. ResNet achieves ROC-AUC of at least $0.8$ across all three datasets when no experimental data is used (training proportion = 0), outperforming the performance of training directly from experimental data consistently as train

size varies from 0.05 to 0.2. Moreover, the performance of physics-augmented models can be further enhanced through transfer learning, which effectively bridges the gap between simulation and experimental data. Particularly, the accuracy of transfer learning is approximately 0.2 and 0.1 higher than training directly from experimental data when training proportion is 0.05 and 0.2, respectively (Figure 4).

## 6.2 NON-HOMOGENEITY INTRODUCES MORE CHALLENGES

The ROC curves (Figure 5) for ResNet were generated using the One-vs-Rest strategy, where a given class is regarded as the positive class and the remaining classes are regarded as the negative class as a bulk. For Titin I27O, the ROC-AUC remains consistently high, above 0.94 in all cases, regardless of whether experimental data is included in training (Figure 5a, Table 5). For both utrophin and dystrophin, the no-molecule class consistently performs well (ROC-AUC is 1.0) whether experimental data is used for training or not. However, significant improvements are observed for the single-molecule and multiple-molecule classes when experimental data is used during training, with ROC-AUC of single-molecule class increasing by approximately 0.2 for both utrophin and dystrophin (Figure 5b-c, Table 5). This suggests notable differences between simulation and experimental data for both utrophin and dystrophin.

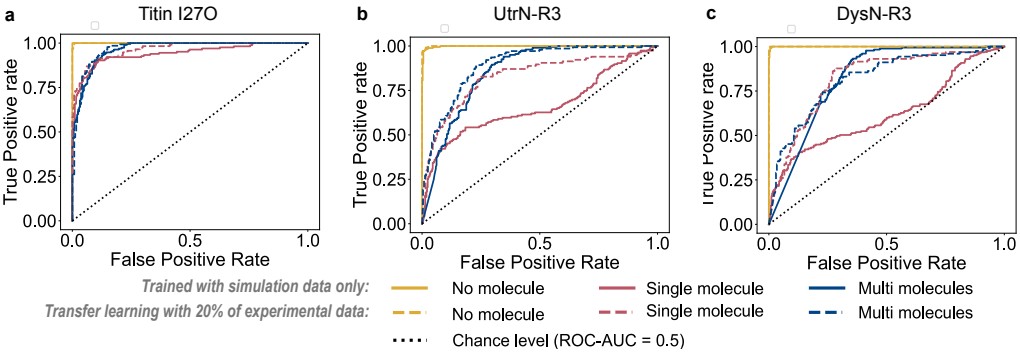

Figure 5: ROC curves from a single run for three datasets, (a) Titin I27O, (b) UtrN-R3, and (c) DysN-R3 using ResNet. ROC curves are plotted using One-vs-Rest strategy, with no molecule, single molecule and multiple molecules classes depicted in yellow, red and blue, respectively. Solid lines represent results trained with simulation data only, while dashed lines indicate results from transfer learning with 20% of experimental data used for training. These ROC curves are generated using 70% of experimental data, with the corresponding ROC-AUC in Table 5.

The ROC curves allow us to adjust the optimal probability threshold. We can achieve high precision but accept low sensitivity to ensure the reliability of single molecule data statistics. The probability threshold can be chosen to tune for a minimum sensitivity or for a minimum specificity. For example, we have selected the thresholds to achieve 97%, 83%, and 82% of the positive identifications of single molecule are correct, for Titin I27O, UtrN-R3, and DysN-R3, respectively. Thus the method can effectively filter data such that inferences can be drawn from true single molecule experiments. Alternatively, we can aim for high sensitivity but accept low precision to avoid losing data, given the scarcity and high cost of producing SMFS data (Appendix C.4).

We further investigated the possible reasons for the superior performances of Titin I27O compared to UtrN-R3 and DysN-R3. Titin I27O has identical domains whereas utrophin and dystrophin have heterogeneous domains. The simulation model we use assumes identical folded domains in a protein. A model that can simulate a molecule with heterogeneous domains requires the model parameters for each domain. These parameters are not available for UtrN-R3 and DysN-R3. Thus, we hypothesize that when information about the molecular domains is missing in the simulation model, the addition of a small batch of experimental data in training provides substantial improvements in performance. This is the case with molecules with heterogeneous domains.

## 7  CONCLUSIONS AND LIMITATIONS

Single-molecule force spectroscopy (SMFS) data of protein molecules are time and resource intensive to collect. Currently, manual visual inspection remains the primary method for classifying force curves resulting from single proteins. These factors make it challenging to obtain precise statistics of single molecular force curves and to generate a large, annotated dataset appropriate for training deep learning models. Consequently, the study of deep learning methods for single-molecule force spectroscopy data is very limited. To address these challenges, we developed an effective classification strategy, where we augment deep learning models with the physics of protein unfolding, enabling accurate detection and classification of SMFS experimental data as originating from no molecules, single molecules, or multiple molecules.

We introduce a physics-based simulation algorithm to generate large and well-annotated data across classes of no molecule, single molecule, and multiple molecules. We also provide experimental datasets, obtained from non-specific single molecule force spectroscopy of three molecules: Titin I27O, UtrN-R3, and DysN-R3. We demonstrate that deep learning models exhibit better performance when pre-trained with our physics-based simulation data than when trained directly using the labeled experimental data. Our physics augmented strategy does not require labeling of the experimental data, avoiding the need for expensive expert adjudication and human bias. In cases where the molecule under study does not have accurate model parameters, we show that the addition of a small amount (approximately $20\%$) supplemental experimental data to the training data and the use of transfer learning, improves average accuracies improved by $6.8\%$.

We identify the following limitations in our study. First, we use the adjudication of a single expert to label the SMFS experimental data used for testing performances. Second, the simulation algorithm relies on energy landscape parameters, which are not straightforward to obtain and typically necessitate experiments involving multiple pulling speeds or constant forces. Transforming our physics augmented classification strategy to be agnostic to the energy landscape parameters is beyond the scope of this study. However, our method does not heavily depend on accurate simulation parameters, as discussed in Section C.5. Third, in our physics-based protein unfolding model, we assume every protein domain behaves identically. However, many proteins, including utrophin and dystrophin have folded domains that are significantly different from each other. Developing methods that can extract the individual energy landscapes of the different dissimilar domains in a protein is likely to aid in identifying single molecule force curves arising from proteins with dissimilar domains. The development of techniques to extract distinct energy landscapes from a small amount of experimental data should be investigated in the future.

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

# A    MORE SIMULATION DETAILS

## A.1    THE DUDKO-HUMMER-SZABO MODEL

Dudko-Hummer-Szabo (DHS) model (Dudko et al., 2008) provided an expression to find the force-dependent transition rate $k_{off}(F)$ by,

$$k_{off}(F) = k_0 \left( 1 - \frac{\nu F \Delta x^{\ddagger}}{\Delta G^{\ddagger}} \right)^{\frac{1}{\nu}-1} e^{\beta \Delta G^{\ddagger} \left[ 1 - \left( 1 - \frac{\nu F \Delta x^{\ddagger}}{\Delta G^{\ddagger}} \right)^{1/\nu} \right]}, \tag{5}$$

where $k_0$ is the intrinsic transition rate, $\Delta x^{\ddagger}$ is the distance to energy barrier, $\Delta G^{\ddagger}$ is the energy barrier height, and $\beta = \frac{1}{k_B T}$, and $\nu = 1/2$ or $2/3$, representing the cusp-like or linear-cubic energy landscape. Here $k_0$, $\Delta x^{\ddagger}$, and $\Delta G^{\ddagger}$ are defined in the absence of external force. The parameters for Titin I27O are reported by (Dudko et al., 2006), while those for UtrN-R3 and DysN-R3 are reported by (Hua et al., 2024).

Table 2: DHS model parameters of three molecules: Titin I27O, UtrN-R3, and DysN-R3

| Molecules | DHS $\nu = 1/2$ | | |
| --- | --- | --- | --- |
| | $ln(k_0)$ | $\Delta x^{\ddagger}$ [nm] | $\Delta G^{\ddagger}$ [$k_B T$] |
| Titin I27O (Dudko et al., 2006) | $-9.210$ | 0.400 | 20 |
| UtrN-R3 (Hua et al., 2024) | $-2.501$ | 0.375 | 9.300 |
| DysN-R3 (Hua et al., 2024) | $-4.501$ | 0.600 | 11.500 |

## A.2 Adhesion force model

The adhesive force can be composed of various components like van der Waals force, capillary force, and chemical forces, which depend on environmental conditions such as roughness, interacting angles, and wetness (Leite et al., 2012; Israelachvili, 2011). However, quantifying these environmental conditions is challenging, and they can vary significantly between experiments. Consequently, we adopt a straightforward yet versatile method to model adhesive force rather than a more intricate approach,

$$F_a(t) = \begin{cases} \frac{t}{t_1} F_{ad} & 0 < t < t_1 \\ \frac{t-t_2}{t_2-t_1} F_{ad} & t_1 < t < t_2 \\ 0 & else \end{cases}, \tag{6}$$

where the adhesive force increases linearly in the interval $[0, t_1]$, reaching the adhesive force threshold $F_{ad}$ at $t_1$, then adhesion between the cantilever tip and the substrate begins to disconnect at $t_1$ and vanishes at $t_2$. The vanishing phase $[t_1, t_2]$ should be much faster than the adhesive phase $[0, t_1]$, with a common choice being to set $t_2$ close to $t_1$, for example, $t_2 = 1.1t_1$. To introduce stochasticity and enhance generality, we assume both $F_{ad}$ and $t_1$ to be Gaussian distributed random variables with user-specified mean and standard deviation.

# B Datasets and deep learning models

## B.1 Deep learning models

**Baselines**  The MLP (Wang et al., 2016; Fawaz et al., 2019) contains three hidden fully connected layers with 500 neurons each, with the RELU activation function. The final layer is a softmax classifier with three neurons, corresponding to the three classes. Dropout rates of 0.1, 0.2, 0.2, and 0.3 are applied to the four layers, respectively. Additionally, a flatten layer is included to reshape the augmented data. The categorical cross-entropy loss is employed in this case,

$$L(x) = -\sum_{j=1}^{K} y_j log(\hat{y}_j), \tag{7}$$

where $x$ is the input time series, $y_j$ is $j$-th component of the one-hot encoding of the true label, and $\hat{y}_j$ is the predicted probability of $x$ belonging to class $j$ out of a total of $K$ classes.

The triplet network (Hoffer and Ailon, 2018) takes three samples: the anchor $x$, the positive sample $x^+$, and the negative sample $x^-$. Here, $x$ is the sample under classification, $x^+$ comes from the same class as $x$, and $x^-$ is of different class. All three samples are passed through the same network architecture, where the weights are shared, to learn representations in the embedding space. The triplet loss is employed,

$$L_{triplet} = max\{0, ||Net(x^+) - Net(x)||_2^2 - ||Net(x^-) - Net(x)||_2^2 + m\}, \tag{8}$$

where $|| \cdot ||$ denotes the $L_2$ norm, and $m$ represents the margin parameter that controls the separation between positive and negative samples in the embedding space. The objective is to minimize the distance between the anchor and the positive sample while maximizing the distance between the anchor and the negative sample in the embedding space. Subsequently, the resulting embeddings are fed into the MLP described earlier to learn classification. The embedding network adopts the ResNet structure, which will be elaborated on below.

**FCN**  FCN (Wang et al., 2016) is composed of three convolutional blocks followed by a Global Average Pooling (GAP) layer and a softmax layer, as illustrated in Figure 2. Each convolutional block includes a 1D convolutional layer, batch normalization, and a Rectified Linear Unit (RELU) activation layer. The output of the final convolutional block is passed through a GAP layer, followed by a fully connected softmax layer. A stride of 1 with zero padding is used to preserve the length of time series data. The filter sizes and kernel sizes of the three convolutional layers are $\{128, 256, 128\}$ and $\{8, 5, 3\}$, respectively.

**ResNet** The ResNet (He et al., 2016) extends neural networks to very deep structures by incorporating shortcut connections. These connections enable the gradient flow directly through the network, easing the training of deeper models. Our Residual Network (ResNet) comprises three residual blocks followed by a GAP layer and a softmax layer. Each residual block contains three convolutional blocks similar to those in the FCN architecture. The output of the last convolutional block is added to the input of the residual block before proceeding to the next layer. The kernel size for the three convolutional layers in each residual block remains consistent with FCN architecture, set as $\{8, 5, 3\}$. The number of filters is the same within the same residual block, with filter sizes specified as $\{64, 128, 128\}$ for three residual blocks, respectively.

**InceptionNet and InceptionTime** InceptionTime ensembles the predictions of five InceptionNet models, each with different initializations, to overcome the high variance across different runs Ismail Fawaz et al. (2020). The InceptionNet model consists of two residual blocks, followed by a global average pooling (GAP) layer and a softmax layer. Each residual block includes three Inception modules instead of fully convolutional layers. The default hyperparameter values are used.

## B.2 MORE DETAILS OF DATASETS

**DDRs** The dataset contains unfolding curves of interaction forces between DDRs and its ligand, collagen, measured by AFM at the single molecule level (Waite et al., 2023). The dataset contains three classes: 1) no molecule, 2) single molecule, and 3)multiple molecules. However, force curve collection was conducted at a single pulling speed, and the dataset is standardized to have equal lengths for each sample.

Figure 6 displays example force data traces for both simulation and experimental data of Titin I27O across three classes. Similarly, Figure 7 and 8 show the corresponding examples for UtrN-R3 and DysN-R3, respectively.

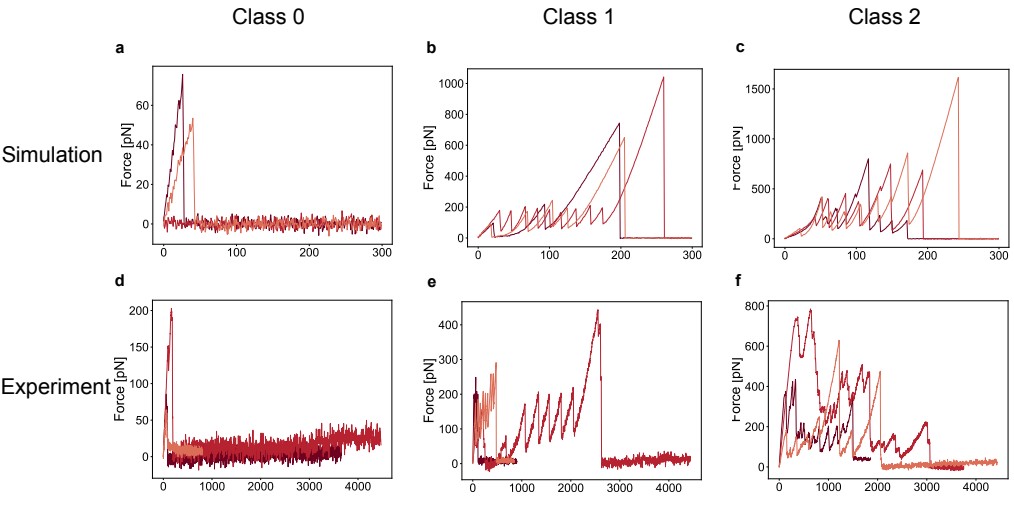

Figure 6: Titin I27O example traces from both simulation (a-c) and experimental (d-f) datasets of all three classes; Class 0 has no protein between the substrate and cantilever tip, Class 1 has one protein, and Class 2 has at least two proteins attached.

## B.3 DATA PRE-PROCESSING

In both single molecule and multiple molecule cases, the force curve typically consists of two parts, separated by the detachment of the event. Our interest lies in the region before detachment, where unfolding events occur. Here, the force curves are trimmed to retain the region before detachment by identifying the detachment point. This trimming process is illustrated in the transition from Figure 9a

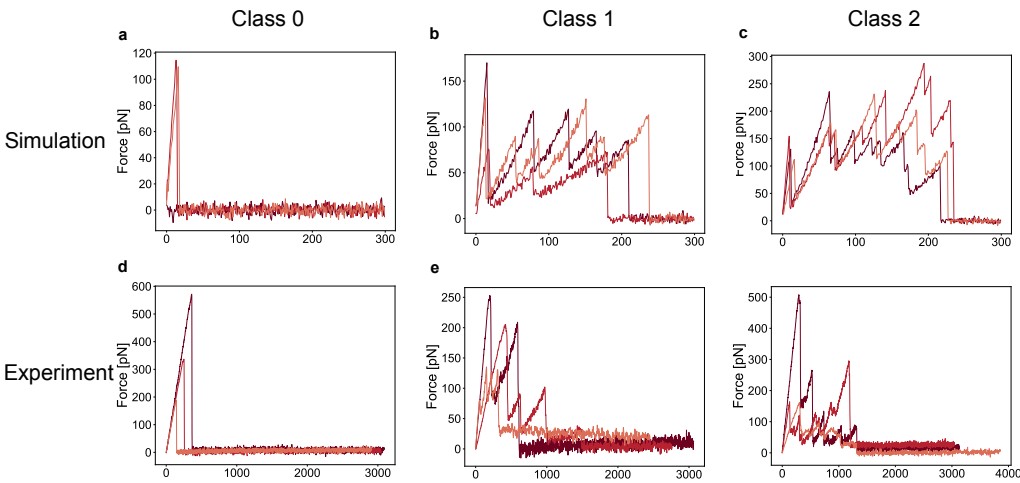

Figure 7: UtrNR3 example traces from both simulation (a-c) and experimental (d-f) datasets of all three classes; Class 0 has no protein between the substrate and cantilever tip, Class 1 has one protein, and Class 2 has at least two proteins attached.

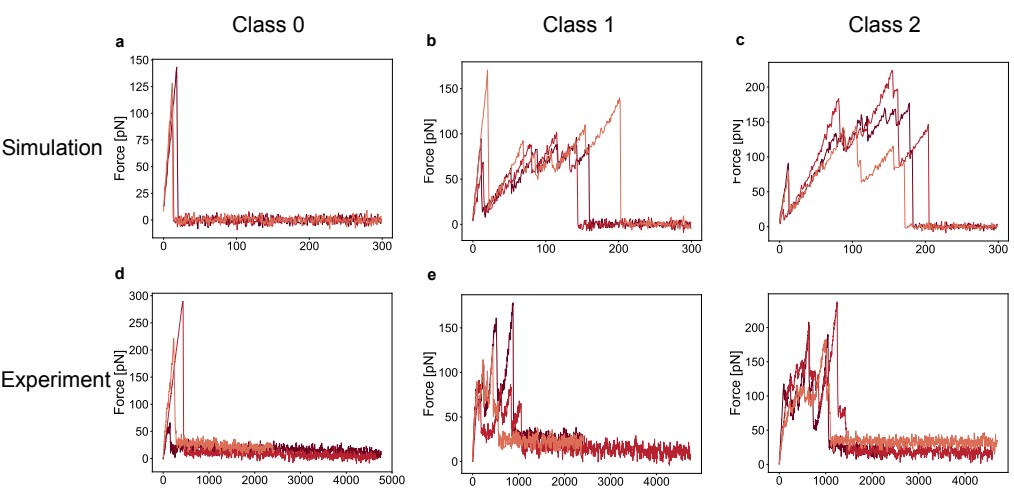

Figure 8: DysNR3 example traces from both simulation (a-c) and experimental (d-f) datasets of all three classes; Class 0 has no protein between the substrate and cantilever tip, Class 1 has one protein, and Class 2 has at least two proteins attached.

to Figure 9b. To address the issue of unequal data length, linear interpolation is employed to resample data to a uniform data length. This step is important since deep learning models require inputs of equal length. Next, we normalize the time series, as depicted in the transition from Figure 9b to Figure 9c, to ensure that the deep learning models focus on learning shapes rather than magnitudes. Magnitudes might be misleading, given their wide range, as discussed in Section 4.2. For normalization, we opt for min-max normalization, $x_{scaled} = \frac{x - x_{min}}{x_{max} - x_{min}}$, where $x_{min}$ and $x_{max}$ are minimum and maximum values of the data $x$. We utilize tools from the publicly available time series library Tslearn (Tavenard et al., 2020) to implement the two preprocessing steps: resampling and normalization.

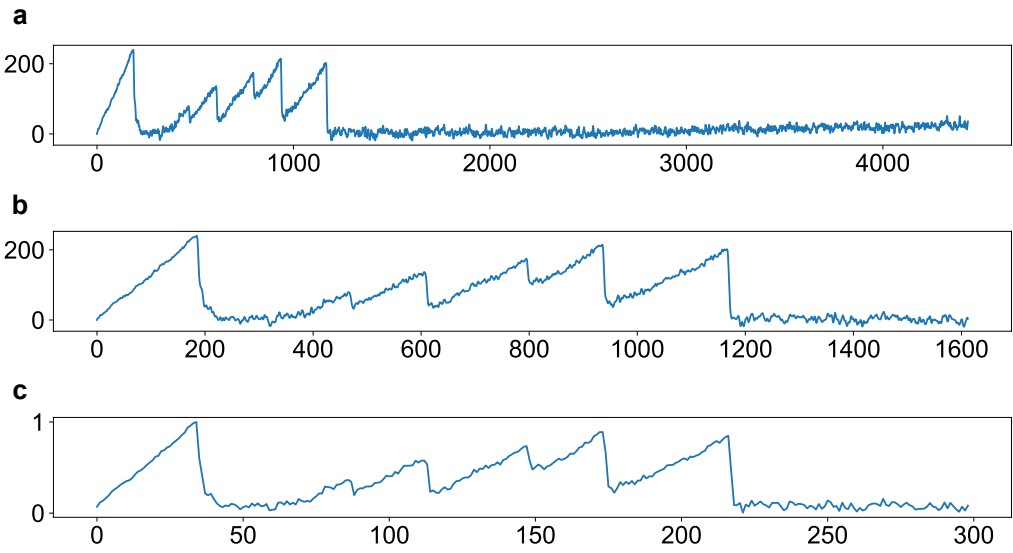

Figure 9: Data preprocessing example. (a) The original force curve. (b) The force curve after trimming. (c) The force curve after normalization.

### B.4 IMPLEMENTATION DETAILS

The MLP is trained with AdaDelta with learning rate $0.1$ and decay rate $0.95$ (Wang et al., 2016). FCN, ResNet, InceptionNet, and InceptionTime are trained with Adam (Kingma and Ba, 2017) with the learning rate $0.001$, $\beta_1 = 0.9$, $\beta_2 = 0.999$ and $\epsilon = 1e - 8$. Triplet is also trained with the same parameters to learn embeddings initially and then trained with the same parameters as MLP to learn classifications. The best-performing model, determined by achieving the lowest training loss, is then selected and evaluated on the experimental data. These models are trained with Apple M1 Pro, which has 10-core CPU and 16-core GPU.

### B.5 SIMILARITIES BETWEEN SIMULATED AND EXPERIMENTAL DATA

The experimental and simulated data are compared using statistically obtained unfolding forces, with a particular focus on the most probable values. The unfolding force statistics originating from single molecule are presented using violin plots for both experimental and simulated data of three protein molecules: Titin I27O, UtrN-R3, and DysN-R3 (Figure 10). These violin plots illustrate data distributions using kernel density estimation, represented as black lines on each side, where the width of each curve indicates the relative frequency of data points. Black stars mark the most probable values, corresponding to the widest sections of the violin plots.

We observe that the most probable values are quite similar between the simulated and experimental data for all three protein molecules. Titin I27O shows the smallest difference at 1 pN, while UtrN-R3 and DysN-R3 exhibit differences of around 10 pN. However, there is a noticeable tail of high-magnitude values in the experimental data, which may result from factors such as variations in sample homogeneity or bond interactions.

## C MORE RESULTS

### C.1 DISCUSSION OF OVERFITTING

We have minimized the risk of overfitting by incorporating techniques such as dropout, global average pooling, and batch normalization into the model (Goodfellow et al., 2016). Further, we evaluate our model performances both on non-overlapping validation data derived from simulations and on

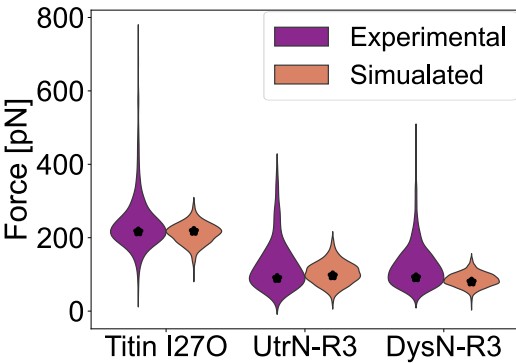

Figure 10: Unfolding force distributions are depicted using violin plots, with black stars indicating the positions of the most probable values. The violins represent data distributions through kernel density estimation, shown as black lines on each side, with the width of each curve reflecting the relative frequency of data points. Experimental and simulated data are visualized in purple and yellow, respectively. The most probable values, from left to right, are 216.35, 217.75, 89.45, 96,65, 91.25, and 79.75 pN.

test data consisting of previously unseen experimental data. We did not observe large deviations in performance metrics between the training, validation, and test evaluations, indicating our risk of overfitting is low.

The results for all deep learning models on simulated validation data from three protein molecules, Titin I27O, UtrN-R3 and DysN-R3, are presented in Figure 11. FCN, ResNet, InceptionNet, and InceptionTime outperform MLP and Triplet, achieving a near-perfect ROC-AUC of 1. This suggests that our pre-trained models are not overfitting to the simulated training data.

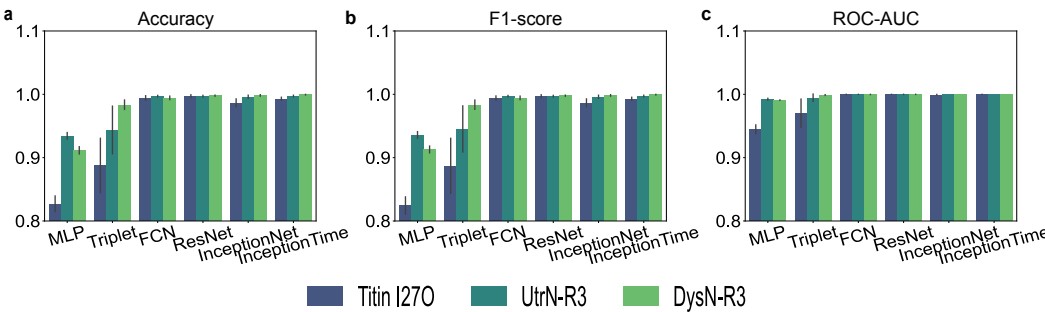

Figure 11: The metrics, including (a) accuracy, (b) F1-score, and (c) ROC-AUC, for deep learning models on simulated validation data from three protein molecules: Titin I27O, UtrN-R3, and DysN-R3. Error bars indicate the standard deviations over five runs. One reference curve ($M = 1$) is used.

## C.2 DDRs DATASET

All deep learning models except triplet were trained using $80\%$ of the total dataset as the training dataset and tested on the remaining dataset. Their performance is compared to that of the triplet model reported in (Waite et al., 2023), as presented in Table 3. The dataset was pre-processed as proposed in (Waite et al., 2023), which involves applying a numerical first-order derivative, followed by a moving average filter with a window size of 13, and finally min-max normalization. Both FCN and ResNet outperform Triplet, with at least $10\%$ higher overall accuracy and F1 score.

Table 3: Overall accuracy, class accuracy, and F1-score of DDRs dataset for Triplet, MLP, FCN, ResNet, InceptionNet, and InceptionTime are presented in the format of average (standard deviation) from 5 runs. Among these models, InceptionNet demonstrates superior performance compared to the others.

| Models | Overall accuracy (%) | Class accuracy (%) | | | F1-score (%) |
|---|---|---|---|---|---|
| | | Class 0 | Class 1 | Class 2 | |
| Triplet (Waite et al., 2023) | 66.70 | 73.30 | 63.30 | 63.30 | 66.70 |
| MLP | 34.84(3.53) | 87.21(17.88) | 10.94(12.49) | 8.24(18.41) | 22.59(8.54) |
| FCN | 77.10(5.15) | 81.22(4.96) | 77.49(13.59) | 73.48(10.06) | 77.30(5.32) |
| ResNet | 79.03(4.84) | 81.90(10.52) | 83.97(4.74) | 70.92(7.44) | 79.02(4.88) |
| **InceptionNet** | 85.48(1.14) | 97.14(2.61) | 84.00(8.94) | 72.50(12.96) | 85.26(1.02) |
| InceptionTime | 77.10(3.50) | 87.97(6.99) | 86.63(4.54) | 57.48(12.95) | 76.45(3.99) |

Table 4: The accuracy, F1-score, and ROC-AUC for the Titin I27O dataset with $M = 3$ reference curves are presented for the FCN and ResNet models. The performance is reported for both the unbalanced and balanced reference curve selection methods, in the format of average (standard deviation) over 5 runs. In the unbalanced criterion, reference curves are randomly selected without regard to class balance, while in the balanced criterion, reference curves are equally selected from each class. The unbalanced criterion slightly outperforms the balanced criterion across all metrics.

| | FCN | | ResNet | |
|---|---|---|---|---|
| | Balanced | Unbalanced | Balanced | Unbalanced |
| Accuracy | 0.8265(0.0366) | 0.8519(0.0140) | 0.7646(0.0556) | 0.8108(0.0347) |
| F1-score | 0.8220(0.0394) | 0.8478(0.0145) | 0.7559(0.0575) | 0.8028(0.0322) |
| ROC-AUC | 0.9460(0.0121) | 0.9494(0.0068) | 0.9187(0.0119) | 0.9203(0.0257) |

## C.3 SIMULATED REFERENCE CURVES IMPROVE PERFORMANCE

We first empirically evaluated two different criteria for selecting reference curves—balanced and unbalanced—using the Titin I27O dataset with $M = 3$ reference curves. FCN and ResNet were selected for evaluation, as they are computationally lighter than InceptionTime while maintaining competitive performance (Figure 3). The balanced criterion involved selecting reference curves equally from each class in the training dataset, while the unbalanced criterion involved selecting reference curves randomly from the entire training dataset without regard to class balance. As shown in Table 4, the unbalanced criterion slightly outperformed the balanced criterion across all metrics. Additionally, the unbalanced method does not require expert annotation, making it more efficient. Therefore, the unbalanced criterion is used as the default method for selecting reference curves unless otherwise specified.

We investigate the impact of the number of reference curves, $M$, on three datasets: Titin I27O, UtrN-R3, and DysN-R3. The deep learning models were pre-trained with simulation data and evaluated on all experimental data. For the Titin I27O dataset, when no reference curve is augmented ($M = 0$), all models perform poorly on experimental data, with an accuracy of less than 0.8 (see Figure 12). Augmenting one or three reference curves ($M = 1$ or $M = 3$) improves all metrics for FCN, ResNet and InceptionNet, while no significant improvement can be observed for MLP and Triplet. Particularly for FCN, the accuracy improves by more than 0.2. As the number of reference curves increases to 5 or 7, the performance decreases. InceptionTime is not included in this analysis for computational efficiency, as it is an ensemble of five InceptionNet models and is expected to yield similar results.

For the UtrN-R3 dataset, FCN, ResNet, and InceptionNet exhibit similar performance across all metrics, outperforming MLP and Triplet (Figure 13). Similarly, with the DysN-R3 dataset, FCN, ResNet and InceptionNet demonstrate good performance, with MLP performing comparably as well (Figure 14). However, the influence of the number of reference curves in both the UtrN-R3 and DysN-R3 datasets is less pronounced compared to the Titin I27O dataset (Figure 12). In conclusion,

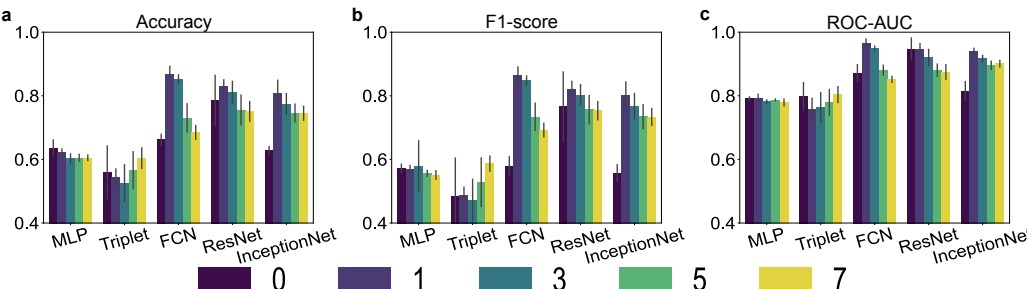

Figure 12: The metrics of deep learning models investigating the impact of the number of reference curves $M$ for Titin I27O experimental data, including (a) accuracy, (b) F1-score, and (c) ROC-AUC. Each metric is plotted against the number of reference curves, which ranges from 0 to 7, with error bars indicating the standard deviations over five runs.

using $M = 1$ as the number of reference curves emerges as the optimal choice, consistently yielding superior performance with the Titin I27O dataset and comparable performance with the UtrN-R3 and DysN-R3 datasets.

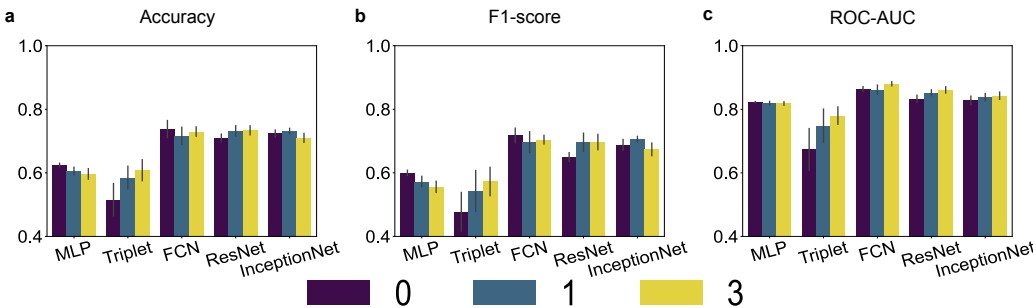

Figure 13: The metrics of deep learning models investigating the impact of the number of reference curves $M$ for UtrN-R3 experimental data, including (a) accuracy, (b) F1-score, and (c) ROC-AUC. Each metric is plotted against the number of reference curves, which ranges from 0 to 3, with error bars indicating the standard deviations over five runs.

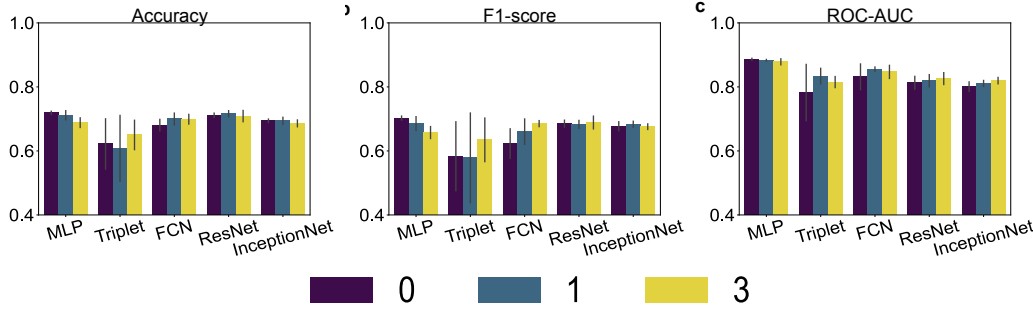

Figure 14: The metrics of deep learning models investigating the impact of the number of reference curves $M$ for DysN-R3 experimental data, including (a) accuracy, (b) F1-score, and (c) ROC-AUC. Each metric is plotted against the number of reference curves, which ranges from 0 to 3, with error bars indicating the standard deviations over five runs.

Table 5: ROC-AUC of ROC curves in Figure 5 (ResNet).

| | Trained with simulation data only | | | Transfer learning with 20% of experimental data | | |
|---|---|---|---|---|---|---|
| | No molecule | Single molecule | Multiple molecules | No molecule | Single molecule | Multiple molecules |
| Titin | 1.00 | 0.94 | 0.96 | 1.00 | 0.97 | 0.97 |
| UtrN-R3 | 1.00 | 0.66 | 0.86 | 1.00 | 0.83 | 0.89 |
| DysN-R3 | 1.00 | 0.62 | 0.82 | 1.00 | 0.82 | 0.82 |

## C.4 ADJUSTING PROBABILITY THRESHOLDS

The primary goal is to identify single molecule force curves, so we simplify the problem from multi-class to binary classification, focusing on distinguishing force curves from single molecules versus those from no molecule and multiple molecules. By leveraging the ROC curve, which illustrates the true positive rates (TPR) against the false positive rates (FPR) by varying the probability threshold $t_p$, we can choose the optimal probability thresholds $t_{op}$ to classify these binary classes. The optimal probability threshold $t_{op}$ is determined by maximizing the difference between TPR and FPR with a weight $\alpha \in \mathbb{R}$ on FPR,

$$t_{op} = argmax_{t_p}(TPR - \alpha \cdot FPR). \tag{9}$$

Subsequently, optimal thresholds with varying $\alpha$ are applied to the binary classification task. The results, compared to the original threshold, are presented in Figure 15 for ResNet. The models are trained on simulation data and then evaluated on experimental data.

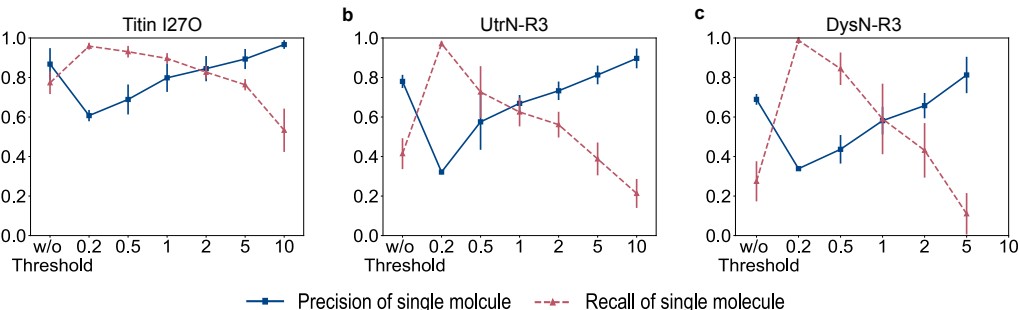

Figure 15: Performance of ResNet with varying probability thresholds for three datasets, (a) Titin I27O, (b) UtrN-R3, and (c) DysN-R3 using ResNet.The x-axis represents values of $\alpha$, with 'w/o Threshold' indicating no threshold is used.

A larger weight $\alpha$ imposes more a greater on false positive, thereby achieving high precision. For example, the precision of single molecule class improved by 0.12 for UtrN-R3 when $\alpha = 10$, 0.1 for Titin I27O when $\alpha = 10$, and 0.1 for DysN-R3 when $\alpha = 5$. However, this improvement comes with a trade-off in recall. This adjustment enhances the reliability of single molecule data statistics, as we can be more confident that no-molecule and multiple-molecules force curves are not misclassified as single-molecule class. Conversely, if the preference is to accept some false positives rather than exclude any true positives as the data are expensive and scarce, $\alpha = 0.2$ or $0.5$ would be the choice to achieve high recall, albeit with lower precision. The scenario where $\alpha = 10$ is omitted for DysN-R3 because the ROC curve exhibits a gradual increase when the false positive rate is low (Figure 5c), causing a negative difference in Equation 9.

## C.5 THE DEGREE OF DEPENDENCE ON ACCURATE SIMULATION PARAMETERS

We intermingle the training and testing data to assess the degree of dependence on accurate simulation parameters. We pre-trained ResNet using simulated data from three different protein molecules: Titin I27O, UtrN-R3, and DysN-R3, each with distinct energy landscape parameters (Table 2). The pre-trained models were then tested on three experimental datasets, with the results summarized in

Table 6: Accuracy, F1-score, and ROC-AUC are recorded for ResNet pre-trained with different simulation datasets across three experimental datasets: Titin I27O, UtrN-R3, and DysN-R3. Results are presented as averages with standard deviations from five runs, and the best performance in each row is bolded.

|  |  | Pre-trained with Titin I27O simulated data | Pre-trained with UtrN-R3 simulated data | Pre-trained with DysN-R3 simulated data | Pre-trained with Titin I27O + UtrN-R3 + DysN-R3 simulated data |
|---|---|---|---|---|---|
| Testing accuracy | Titin I27O | 0.8287(0.0217) | 0.7724(0.0358) | 0.7593(0.0165) | **0.8716(0.0189)** |
|  | UtrN-R3 | 0.5682(0.0784) | **0.7320(0.0163)** | 0.6680(0.0072) | 0.7094(0.0102) |
|  | DysN-R3 | 0.6090(0.0523) | 0.6770(0.0120) | **0.7164(0.0093)** | 0.7034(0.0077) |
| Testing F1-score | Titin I27O | 0.8221(0.0234) | 0.7534(0.0504) | 0.7459(0.0253) | **0.8594(0.0223)** |
|  | UtrN-R3 | 0.4808(0.0756) | **0.6962(0.0288)** | 0.6504(0.0100) | 0.6757(0.0139) |
|  | DysN-R3 | 0.5029(0.0549) | 0.6107(0.0133) | **0.6827(0.0126)** | 0.6550(0.0173) |
| Testing ROC-AUC | Titin I27O | 0.9474(0.0167) | 0.9146(0.0145) | 0.8960(0.0146) | **0.9527(0.0076)** |
|  | UtrN-R3 | 0.7987(0.0457) | **0.8522(0.0090)** | 0.7664(0.0147) | 0.8444(0.0069) |
|  | DysN-R3 | 0.7941(0.0535) | 0.7753(0.0164) | 0.8194(0.0190) | **0.8301(0.0189)** |

Table 6. Although there is a performance drop when the training and testing datasets are mismatched, the decrease is minimal, with a maximum reduction of 0.06 in ROC-AUC. Furthermore, by expanding the pre-training dataset to include simulated data from all three molecules,Titin I27O, UtrN-R3, and DysN-R3, we achieve performance comparable to that obtained with accurate simulation parameters. Additionally, the ROC-AUC for Titin I27O and DysN-R3 improves further by 0.05 and 0.1, respectively, as shown in the last column of Table 6.

