# OpenReview forum: "Identifying single molecule force spectroscopy data using deep learning with physics augmentation"
_ICLR.cc/2025/Conference — ICLR 2025 Conference Withdrawn Submission_

### Official Review · Reviewer_zb6n · 2024-10-18

**Soundness:** 4
**Presentation:** 4
**Contribution:** 4
**Rating:** 1
**Confidence:** 1

**Summary:**

While I recognize the potential significance of the work in understanding protein folding and unfolding using Single Molecule Force Spectroscopy (SMFS) data, I must inform AC and SAC that this specific area is outside my main expertise. Given this, I may not be able to offer detailed or technically relevant feedback that would benefit the review process.

I am happy to provide general comments regarding the manuscript's clarity and structure if necessary, but I wanted to notify you in case you would prefer to reassign this review to someone with more specialized knowledge in SMFS and its related methodologies.

Thank you for your understanding.

**Strengths:**

None

**Weaknesses:**

None

**Questions:**

None

---

> ### Author Response · Authors · 2024-11-21
> **Response to Reviewer zb6n**
>
> Thank you for your time and for reviewing the non-technical aspects of our work as ‘excellent’ (Soundness, Presentation, Contribution scores of ‘4: excellent’). We wish to bring to your attention the overall score of ‘1: strong reject’, which seems in conflict with your intention. We are happy to address any concerns you have about our paper. If there are no concerns, we would appreciate your consideration in raising our overall score.

---

> > ### Comment · Reviewer_zb6n · 2024-11-21
> >
> > I think you don't need to worry about my score, as I have stated clearly that I am unfamiliar with your task and area. AC will definitely attach little importance to my comment.
> >
> > However, after a careful look at other reviewers' comments, I agree with Mv6G's point that your work's main contribution lies in introducing a new dataset and applying mature ML methods to it. In Figure 5, simple CNNs can achieve pretty high AUC-ROC, demonstrating it is not a big challenge. Have you considered proposing a more advanced framework to tackle the problem? Besides, what is the difference between single molecule force spectroscopy and molecular dynamics simulations?

---

> > > ### Author Response · Authors · 2024-11-25
> > >
> > > *Figure 5 does not show the performance of simple CNNs applied on the experimental data*. We have indeed proposed a physics-based framework for tackling the problem of classifying the single molecule force spectroscopy (SMFS) force curves. No existing literature on SMFS has used this approach [1, 2]. The results of this new physics-based approach are shown in Figure 5. While the performance for detecting the ’no-molecule’ case is high, detecting the ‘single molecule’ and ‘multiple molecules’ cases, especially for native human proteins such as utrophin (Figure 5b) and dystrophin (Figure 5c), remains a challenging problem that necessitates our framework.
> > >
> > > Our framework demonstrates superior performance (see Figure 4 a, b, and c) compared to a simplified black-box classification model trained solely on experimental data. Besides the higher performance, we highlight the following significant advantages of our strategy. First, SMFS is an experimental technique often applied on molecules that have not been previously characterized using force spectroscopy. Therefore, labeling their force curves is difficult and fraught with human biases. Our strategy of using a simulation engine to generate the training data, where the ground truth (i.e. labels) are known avoids this issue. Second, the experimental data is subject to experimental conditions (e.g. temperature and unfolding rate). Our physics based framework can train classification models that account for these changes, unlike a classification model that uses only experimental data. Third, SMFS experiments are expensive in both time and resources. Thus, experimental training data on new proteins are hard to generate. Our strategy overcomes such a dearth of training data by pretraining with physics-based simulation data.
> > >
> > > Single-molecule force spectroscopy is a *real physical experimental* technique wherein an Atomic Force Microscope (AFM) is used to manipulate and measure real forces from real physical protein molecules [3]. Molecular dynamic simulations are a complementary technique that can be used to study biomolecules but are heavily reliant on model approximations and computationally expensive [4]. On the other hand, SMFS generates direct experimental data about the physics of proteins without relying on computational approximations.
> > >
> > > While our current work establishes a robust framework for automated classification of no molecule, single molecule, and multiple molecules, this is just the beginning. We aim to further refine the model’s accuracy and extend its applicability to better understand protein heterogeneity in future studies. Our goal is to develop a machine learning application to automate SMFS-related research, alleviating the need for labor-intensive and tedious visual inspection.
> > >
> > > [1] Waite, Joshua R., et al. "Few-shot deep learning for AFM force curve characterization of single-molecule interactions." Patterns 4.1 (2023).
> > >
> > > [2] Doffini, Vanni, et al. "Iterative Machine Learning for Classification and Discovery of Single-Molecule Unfolding Trajectories from Force Spectroscopy Data." Nano Letters 23.22 (2023): 10406-10413.
> > >
> > > [3] Bustamante, Carlos, and Shannon Yan. "The development of single molecule force spectroscopy: from polymer biophysics to molecular machines." Quarterly Reviews of Biophysics 55 (2022): e9.
> > >
> > > [4] Hollingsworth, Scott A., and Ron O. Dror. "Molecular dynamics simulation for all." Neuron 99, no. 6 (2018): 1129-1143.

---

> > > > ### Comment · Reviewer_zb6n · 2024-11-25
> > > >
> > > > Thanks for your response. I get a better understanding of single-molecule force spectroscopy. Good luck.

---

### Official Review · Reviewer_fJLp · 2024-10-31

**Soundness:** 2
**Presentation:** 3
**Contribution:** 2
**Rating:** 5
**Confidence:** 3

**Summary:**

The paper presents a method for classifying single-molecule force spectroscopy (SMFS) data using deep learning models enhanced by physics-based simulations. These models classify force curves as originating from no molecule, a single molecule, or multiple molecules, a task traditionally handled through time-consuming expert visual inspection. The authors propose a Monte Carlo simulation framework to create annotated datasets that mimic real-world experimental data, which they use to pre-train deep learning models. The model accuracy improves further with transfer learning on a small subset of experimental data, achieving up to 75.3% accuracy and a ROC-AUC of 0.87 across multiple datasets.

**Strengths:**

- Originality
  - The originality arises from the proposed physics-augmented simulation framework that enables deep learning models to learn realistic representations of SMFS force curves without relying heavily on annotated experimental data.
- Quality
  - Extensive comparative analysis across deep learning architectures was conducted.
  - The proposed method outperforms previous works.
- Clarity
  - The method is described clearly.
- Significance
  - This study offers a generalizable solution for SMFS data classification, a critical need in the study of protein folding and unfolding mechanisms.
  - The reduction in dependency on expert visual inspection and annotated data also lowers barriers to adoption, making SMFS analysis more accessible to the broader biological research community.

**Weaknesses:**

- Assumptions in simulation
  - The Monte Carlo (MC) simulation framework assumes identical protein domains across molecules, which could limit the model’s effectiveness for proteins with heterogeneous domains. This assumption may restrict the model’s generalizability, particularly in cases where protein unfolding behavior varies between domains.
- Computational efficiency and scalability
  - MC requires considerable computation for each force curve, which might hinder its scalability, particularly in high-throughput SMFS applications that involve thousands of force curves. While this approach is effective in current datasets, its feasibility for larger datasets remains uncertain.

**Questions:**

- Impact of Transfer Learning on Model Performance
  - The paper notes that transfer learning with a subset of experimental data improves model accuracy and ROC-AUC. Could you clarify the minimum amount of experimental data required to achieve significant performance gains?
- Data Diversity and Experimental Validation
  - The experimental datasets focus on non-specific pulling for three proteins. Have you considered validating the model on additional proteins or experimental setups (e.g., different solution conditions or functionalized probes)? If not, do you anticipate any limitations when applying this model to other setups?
- Sensitivity to Simulation Parameters
  - How sensitive is the model to variations in key simulation parameters, such as those defined by the DHS model? Does the accuracy vary significantly if the estimated parameters are slightly off?

---

> ### Author Response · Authors · 2024-11-21
> **Response to Reviewer fJLp comments: on model parameters and scalability**
>
> Thank you for your thoughtful feedback. We have addressed your questions and concerns below. If there are any remaining issues, we would be happy to discuss them further. If there are no additional concerns, we would appreciate your consideration in raising our score.
>
> **Reviewer’s comments**:
>
> *W1.  Assumptions in simulation*
>
> The Monte Carlo (MC) simulation framework assumes identical protein domains across molecules, which could limit the model’s effectiveness for proteins with heterogeneous domains. This assumption may restrict the model’s generalizability, particularly in cases where protein unfolding behavior varies between domains.
>
> **Authors response**:
>
> A *significant* insight of the study is that we can perform the task of classification of data under no molecule, single molecule, and multiple molecules (for proteins with heterogenous domains) being pulled where the *training* is done based on simulation data, where a single double-well potential model of the domains is employed.  This implies that there is enough other information on the force-curves that allows discrimination between no, single, and multiple proteins-based force-curves.
>
> We emphasize that neural networks trained on simulated data using homogeneous domains have the capability to classify between the number of proteins being involved in the experiments of protein molecules with heterogeneous domains. In Figure 4 b and 4 c on page 8, our approach achieves AUCs of 0.86 and 0.83 for the classification task when applied to data from the *heterogeneous proteins* utrophin and dystrophin, respectively. We also show that incorporating a small subset of experimental data ($\sim $100 examples) through transfer learning improves accuracy by 6.8\% and ROC-AUC by 0.06 (Figure 4 of our paper). We would like to clarify that finding parameters specific to each domain of the protein is challenging.
>
> **Reviewer’s comments**:
>
> *W2. Computational efficiency and scalability*
>
> MC requires considerable computation for each force curve, which might hinder its scalability, particularly in high-throughput SMFS applications that involve thousands of force curves. While this approach is effective in current datasets, its feasibility for larger datasets remains uncertain.
> Authors response:
>
> Our physics-based Monte Carlo (MC) simulation algorithm can generate thousands of force curves for each protein within hours (approximately 2100 per hour). This is with the prototype algorithm implemented with a CPU. For scalability, we can leverage GPUs since the simulation instances are independent and can be run in parallel. Furthermore, this algorithm has already been successfully employed in studies investigating the mechanical properties of utrophin and dystrophin [1].
>
> [1] Cailong Hua, Rebecca A. Slick, Joseph Vavra, Joseph M. Muretta, James M. Ervasti, and Murti V. Salapaka. Two operational modes of atomic force microscopy reveal similar mechanical properties for homologous regions of dystrophin and utrophin, May 2024. URL https://www.biorxiv. org/content/10.1101/2024.05.18.593686v1. Pages: 2024.05.18.593686 Section: New Results.

---

> ### Author Response · Authors · 2024-11-21
> **Response to Reviewer fJLp questions**
>
> Thank you for your thoughtful feedback. We have addressed your questions and concerns below. If there are any remaining issues, we would be happy to discuss them further. If there are no additional concerns, we would appreciate your consideration in raising our score.
>
> *Q1. Impact of Transfer Learning on Model Performance*
>
> *The paper notes that transfer learning with a subset of experimental data improves model accuracy and ROC-AUC. Could you clarify the minimum amount of experimental data required to achieve significant performance gains?*
>
> **Authors response**:
>
> Incorporating $\sim$50 experimental force curves via transfer learning leads to an average improvement in accuracy by 2.2\% and in ROC-AUC by 0.02 across three experimental datasets. When $\sim$100 experimental force curves are used, the accuracy improves by 6.8\% and ROC-AUC increases by 0.06.
>
> *Q2. Data Diversity and Experimental Validation*
>
> *The experimental datasets focus on non-specific pulling for three proteins. Have you considered validating the model on additional proteins or experimental setups (e.g., different solution conditions or functionalized probes)? If not, do you anticipate any limitations when applying this model to other setups?*
>
> **Authors response**:
>
> We are utilizing experimental data collected by biochemists [1,2,3], ensuring that the chosen medium is relevant and validated by experts in the field. Additionally, we evaluated our model on a publicly available SMFS dataset of DDR proteins [4], as detailed in Section C.1 of our paper (page 18).
>
> Furthermore, the proteins included in our study, utrophin and dystrophin, are diverse proteins (Figure 1 of [2]). While our current work establishes a robust framework for automated classification of no molecule, single molecule, and multiple molecules, this is just the beginning. We aim to further refine the model’s accuracy and extend its applicability to better understand protein heterogeneity in future studies.
>
> [1] Rajaganapathy, Sivaraman, et al. "Distinct mechanical properties in homologous spectrin-like repeats of utrophin." Scientific reports 9.1 (2019): 5210.
>
> [2] Ramirez, Maria Paz, et al. "Phosphorylation alters the mechanical stiffness of a model fragment of the dystrophin homologue utrophin." Journal of Biological Chemistry 299.2 (2023).
>
> [3] Cailong Hua, Rebecca A. Slick, Joseph Vavra, Joseph M. Muretta, James M. Ervasti, and Murti V. Salapaka. Two operational modes of atomic force microscopy reveal similar mechanical properties for homologous regions of dystrophin and utrophin, May 2024. URL https://www.biorxiv. org/content/10.1101/2024.05.18.593686v1. Pages: 2024.05.18.593686 Section: New Results.
>
> [4] Waite, Joshua R., et al. "Few-shot deep learning for AFM force curve characterization of single-molecule interactions." Patterns 4.1 (2023).

---

> > ### Author Response · Authors · 2024-11-26
> >
> > Dear Reviewer,
> >
> > Thank you for taking the time and effort to review our paper again. We have carefully addressed your valuable comments in detail. As the last day to upload a revised PDF approaches its conclusion, we warmly invite any further feedback or suggestions you might have.

---

> > > ### Comment · Reviewer_fJLp · 2024-11-27
> > >
> > > Thanks for your reply. I have read the response carefully. However, the technical novelty of the approach is limited and I worry about the real application of this research.

---

> > > > ### Author Response · Authors · 2024-11-29
> > > >
> > > > This is the *first work* to use a physics-based framework for classifying SMFS force curves. To our knowledge, no prior literature has applied this methodology in SMFS studies [1,2]. Our goal is to create a machine learning solution that automates SMFS-related research, thereby reducing reliance on expert knowledge and eliminating the labor-intensive, time-consuming process of manual visual inspection.
> > > >
> > > > We have validated our approach using force curves obtained from *real physical experiments* conducted via Atomic Force Microscopy on real protein samples. Dystrophin and utrophin are two real human proteins that have been extensively studied due to their biological significance. Deficiencies of dystrophin lead to severe muscle wasting disorders like Duchenne muscular dystrophy (DMD), a fatal disease occurring in 1 out of 4000 male births [3]. Utrophin, a fetal homologue of dystrophin, is under active investigation as a protein replacement therapy for DMD [4].
> > > >
> > > > Many laboratories have developed heuristic methods to analyze single molecule force spectroscopy (SMFS) data, as seen in [5-10]. Although these methods can perform automated data analysis, they often provide approximate results and require manual adjustments by experts to fine-tune their parameters. In our research group, we aim to utilize this machine learning application to generate more accurate and reliable statistics from SMFS data, particularly for studying the mechanical properties of dystrophin and utrophin in future experiments.
> > > >
> > > > Our method is robust to variations in simulation parameters. To evaluate this, we intermingle the training and testing data to assess the degree of dependence on accurate simulation parameters. Although there is a performance drop when the training and testing datasets are mismatched, the decrease is minimal, with a maximum reduction of 0.06 in ROC-AUC. A more detailed discussion is available in Section C.5 (Page 21) of our revised paper. Furthermore, reporting these simulation parameters is becoming increasingly common when studying new protein molecules. Several examples can be found in recent studies [11-15]. We believe that our strategy of using a simulation engine to generate training data has broad applicability and can significantly aid in automating the classification of SMFS data.
> > > >
> > > > [1] Waite, Joshua R., et al. "Few-shot deep learning for AFM force curve characterization of single-molecule interactions." Patterns 4.1 (2023).
> > > >
> > > > [2] Doffini, Vanni, et al. "Iterative Machine Learning for Classification and Discovery of Single-Molecule Unfolding Trajectories from Force Spectroscopy Data." Nano Letters 23.22 (2023): 10406-10413.
> > > >
> > > >
> > > > [3] Mendell, Jerry R., et al. "Evidence‐based path to newborn screening for Duchenne muscular dystrophy." Annals of neurology 71.3 (2012): 304-313.
> > > >
> > > > [4] Guiraud, Simon, et al. "Advances in genetic therapeutic strategies for Duchenne muscular dystrophy." Experimental physiology 100.12 (2015): 1458-1467.
> > > >
> > > > [5]  Rajaganapathy, Sivaraman, et al. "Distinct mechanical properties in homologous spectrin-like repeats of utrophin." Scientific reports 9.1 (2019): 5210.
> > > >
> > > > [6] Ramirez, Maria Paz, et al. "Phosphorylation alters the mechanical stiffness of a model fragment of the dystrophin homologue utrophin." Journal of Biological Chemistry 299.2 (2023).
> > > >
> > > > [7] Ott, Wolfgang, et al. "Single-molecule force spectroscopy on polyproteins and receptor–ligand complexes: The current toolbox." Journal of structural biology 197.1 (2017): 3-12.
> > > >
> > > > [8] Liu, Zhaowei, et al. "Engineering an artificial catch bond using mechanical anisotropy." Nature Communications 15.1 (2024): 3019.
> > > >
> > > > [9] Jiao, Junyi, et al. "Single-molecule protein folding experiments using high-precision optical tweezers." Optical Tweezers: Methods and Protocols (2017): 357-390
> > > >
> > > > [10] Bustamante, Carlos, et al. "Single-molecule studies of protein folding with optical tweezers." Annual review of biochemistry 89.1 (2020): 443-470.
> > > >
> > > > [11] Hane, Francis T., Simon J. Attwood, and Zoya Leonenko. "Comparison of three competing dynamic force spectroscopy models to study binding forces of amyloid-β (1–42)." Soft matter 10.12 (2014): 1924-1930.
> > > >
> > > > [12] Schoeler, Constantin, et al. "Mapping mechanical force propagation through biomolecular complexes." Nano letters 15.11 (2015): 7370-7376.
> > > >
> > > > [13] Milles, Lukas F., et al. "Molecular mechanism of extreme mechanostability in a pathogen adhesin." Science 359.6383 (2018): 1527-1533.
> > > >
> > > > [14] ​​Liu, Zhaowei, et al. "High force catch bond mechanism of bacterial adhesion in the human gut." Nature communications 11.1 (2020): 4321.
> > > >
> > > > [15] Bustamante, Carlos J., et al. "Optical tweezers in single-molecule biophysics." Nature Reviews Methods Primers 1.1 (2021): 25.

---

### Official Review · Reviewer_Mv6G · 2024-11-03

**Soundness:** 4
**Presentation:** 3
**Contribution:** 3
**Rating:** 8
**Confidence:** 3

**Summary:**

The paper trains a classifier on time vs. force curves from protein unfolding force measurements. For such measurements, the model classifies whether the measurment came from the type of interaction that is desired to be observed, or from other artifacts that should not be included in the assessment of the protein unfolding forces. This has been done before. The application is useful because practitioners can automatically determine which measurments to include in downstream analyses instead of deciding manually. The papers main novelty is a dataset of simulated protein unfolding measurments which are used to train a better classifier next to training on experimental data. The paper is evaluated on three protein's unfolding measurmeents.

**Strengths:**

1. The paper introduces an interesting task to the ML conference community (although it is not the first one to address the specific task with ML solutions).
2. Very good explanations of the new application area.
3. Good experimental protocols that  ensure statistical significance of the results.
4. The paper demonstrates that for these measurments, training on synthetic data suffices for obtaining a generalizable classifier that can predict whether force measurements

**Weaknesses:**

1. Novelty of the application: The paper applies the most basic ML solution to a task. It can still be valuable to introduce a new task to the ML community. The task has been addressed with similar ML tools in previous works. Thus, there is no novelty in application. There is some technical novelty in the creation of a synthetic dataset.
2. I am not completely sure whether this is a weakness since I am not familiar with the task: I would imagine that for most proteins of interest we do not have the same sequence and structure repeating multiple times and then observe the same unfolding pattern of essentially the same protein. Why is there no evaluation for proteins that are not repeating or is this the case for one of your 3 experiments? If this is indeed of interest, and we cannot simply combine a single protein of interest into a chain of repeating proteins, then it seems to me that the evaluations miss the important evaluation and only an easier task is evaluated. The task is easier because classifying whether the same unfolding event and the same pattern occurs in the curve multiple times is easy compared to classifying a single unfolding event.

Minor:
1. The task is very easy. Correct me if I am wrong, but it seems to me that the model could simply classify whether or not there is a repeated pattern in the response measurement.

**Questions:**

1. Are the molcules attached to something when we "pull" on them?
2. As far as I understand, we pull the proteins and observe their unfolding - how are the proteins attached to the head with which we pull?
3. How do we make sure that he molecules are always attached to the pulling head at the same residue?
4. Does the medium in which we measure the pulling force affect the force measurement outcomes?
5. Why would we have the same protein repeated multiple times in a chain? Are they attached together in some fashion?

---

> ### Author Response · Authors · 2024-11-21
> **Response to Reviewer Mv6G: On the novelty and impact**
>
> Thank you for your thoughtful feedback. We have addressed your questions and concerns below. If there are any remaining issues, we would be happy to discuss them further. If there are no additional concerns, we would appreciate your consideration in raising our score.
>
> **Reviewer's Comments**:
>
> *W1. Novelty of the application: The paper applies the most basic ML solution to a task. It can still be valuable to introduce a new task to the ML community. The task has been addressed with similar ML tools in previous works. Thus, there is no novelty in application. There is some technical novelty in the creation of a synthetic dataset.*
>
> **Authors response**:
>
> This is the *first work* to apply deep learning to single molecule force spectroscopy (SMFS) data from non-specific pulling, whereas the previous works are limited to specific pulling SMFS data [1,2]. Specific pulling relies on functionalizing cantilevers, which is time-consuming and requires careful handling and practice. Specific pulling also uses molecular fingerprints, which need to be introduced into the protein of interest that is being investigated. These fingerprints have a signature pattern when unfolding that provides greater discernability in classifying the number of proteins involved in data. However, the approach based on specific pulling requires significant domain expertise; moreover, fingerprints introduce other confounding effects, as these added domains do not exist in the native protein. In contrast, non-specific pulling conducts experiments without functionalizing probes and introducing fingerprints. Thus, non-specific pulling is a more prevalent and easier method in SMFS studies.  Without molecular fingerprints, it is much harder to classify data into categories originating from no molecules, single molecules, or multiple molecules, making our task more challenging. We assert that, to the best of our knowledge, there is no prior report of classifying SMFS data that results from non-specific pulling. The resulting automation will prove to be impactful to the large community of researchers using SMFS (including our group).
>
> SMFS data of protein molecules are time and resource intensive to collect. Currently, manual visual inspection remains the primary method for classifying force curves resulting from single proteins. These factors make it challenging to obtain precise statistics of single molecular force curves and to generate a large, annotated dataset appropriate for training deep learning models. Considerable domain expertise is required to understand physics behind this application to SMFS. Automating the classification of SMFS data to draw influence without expert knowledge is a major contribution of our work.
>
> We agree our focus is not on developing new ML methods; however, our work is within the scope of ILCR, which includes impactful applications of ML. Our work has employed current state-of-the-art deep learning models for univariate time series classification, including fully connected neural networks (FCN), residual networks (ResNet), and InceptionTime. Please let us know if we are missing any relevant ML methods which may lead to better evaluation. We hope we have convinced the reviewer about the novelty of the article and its impact
>
> [1] Waite, Joshua R., et al. "Few-shot deep learning for AFM force curve characterization of single-molecule interactions." Patterns 4.1 (2023).
>
> [2] Doffini, Vanni, et al. "Iterative Machine Learning for Classification and Discovery of Single-Molecule Unfolding Trajectories from Force Spectroscopy Data." Nano Letters 23.22 (2023): 10406-10413.

---

> > ### Author Response · Authors · 2024-11-21
> > **Response to Reviewer Mv6G: On heterogeneity of domains in a protein**
> >
> > **Reviewer's Comments**:
> >
> > *W2. I am not completely sure whether this is a weakness since I am not familiar with the task: I would imagine that for most proteins of interest we do not have the same sequence and structure repeating multiple times and then observe the same unfolding pattern of essentially the same protein. Why is there no evaluation for proteins that are not repeating or is this the case for one of your 3 experiments? If this is indeed of interest, and we cannot simply combine a single protein of interest into a chain of repeating proteins, then it seems to me that the evaluations miss the important evaluation and only an easier task is evaluated. The task is easier because classifying whether the same unfolding event and the same pattern occurs in the curve multiple times is easy compared to classifying a single unfolding event.*
> >
> > **Authors response**:
> >
> > Our primary focus in the study and related evaluations are for proteins with non-repeating domains.  Indeed, of the proteins employed in our tests, only Titin has a repeated structure, and it is used for calibrating and validating our methods. However, utrophin and dystrophin are real human proteins with considerable variations in their sequence and structure. Dystrophin is a protein expressed primarily at the muscle cell membrane, or sarcolemma, in striated muscle tissue.  Deficiencies of this protein lead to severe muscle wasting disorder like Duchenne muscular dystrophy (DMD), a fatal disease occurring in 1 out of 4000 male births [1]. Utrophin is a fetal homologue of dystrophin and is under active investigation as a dystrophin replacement therapy for DMD. Thus, both these proteins are important and are being researched heavily.
> >
> >
> > In all these SMFS studies, the cantilever probes and proteins are too small to be seen by the naked eye. Moreover, as emphasized earlier, multiple proteins can adhere to the cantilever probe, confounding data that are collected, which should be limited to experiments resulting from a single protein. These challenges are further exacerbated by other environmental and experimental factors, such as the concentration of the protein in the buffer solution and the noise and uncertainty of the probing system. Thus, typically for statistically relevant inferences, data from thousands of force curves on a single protein is required, which results from multiple thousands of force curves (which needs to be filtered from data that includes multiple proteins). These challenges remain even when the protein has the same domain repeated multiple times. The task of investigating the protein with heterogenous domains is considerably more difficult. We hope that we have clarified the difficulty of the task.
> >
> > A *significant* insight of the study is that we can perform the task of classification of data under no molecule, single molecule, and multiple molecules (for proteins with heterogenous domains) being pulled where the *training* is done based on simulation data, where a single double-well potential model of the domains is employed.  This implies that there is enough other information on the force-curves that allows discrimination between no, single, and multiple proteins-based force-curves.
> >
> > We emphasize that neural networks trained on simulated data using homogeneous domains have the capability to classify between the number of proteins being involved in the experiments of protein molecules with heterogeneous domains. We also show that incorporating a small subset of experimental data (∼ 100 examples) through transfer learning improves accuracy by 6.8% and ROC-AUC by 0.06.
> >
> > [1] Mendell, Jerry R., et al. "Evidence‐based path to newborn screening for Duchenne muscular dystrophy." Annals of neurology 71.3 (2012): 304-313.

---

> > > ### Author Response · Authors · 2024-11-21
> > > **Response to Reviewer Mv6G M1: On challenges of the task**
> > >
> > > **Reviewer's comments**:
> > >
> > > *M1. The task is very easy. Correct me if I am wrong, but it seems to me that the model could simply classify whether or not there is a repeated pattern in the response measurement.*
> > >
> > > **Authors response**:
> > >
> > > Typically, there is no repeated pattern that is obvious, to isolate data resulting from single protein pulling or multiple protein molecules being pulled. We show through our experimental results in Figure 4 that this task is not an easy one. A classification model trained only on the experimentally observed force curves, without the use of a physics model has a significantly lower performance compared to our approach (see Figure 4 which compares performance metrics with and without physics). Multiple challenges exist in the task – low signal to noise ratios due to the compounded effects of instrument measurement noise and inherent thermal noise in the systems, the uncertainty on which domains of the protein will unfold in a given experiment, the stochasticity of the unfolding forces to name a few.
> > >
> > > The current widely accepted approach relies on visual inspection but is extremely time-consuming and requires significant expertise [1, 2]. Substantial effort is needed to identify and isolate data corresponding to single-molecule events from thousands of traces. Our automated method addresses this challenge by significantly reducing the reliance on manual inspection, paving the way for faster and more consistent analysis of SMFS data. We would like to emphasize that our research group is investigating the mechanical properties of dystrophin and utrophin; the ML application reported here was motivated by the large effort needed to filter the non-admissible data resulting from our experiments that involve multiple-proteins (in contrast to single protein). We expect the impact to be considerable on SFMS related research wherein laborious and tedious visual inspection can be automated.
> > >
> > > [1] Bornschlögl, T., & Rief, M. (2011). Single-molecule protein unfolding and refolding using atomic force microscopy. Single Molecule Analysis: Methods and Protocols, 233-250.
> > >
> > > [2] Ares, Pablo, Julio Gomez-Herrero, and Fernando Moreno-Herrero. "High-resolution atomic force microscopy imaging of nucleic acids." Nanoscale Imaging: Methods and Protocols (2018): 3-17.

---

> ### Author Response · Authors · 2024-11-21
> **Response to Reviewer Mv6G Questions**
>
> *Q1. Are the molecules attached to something when we "pull" on them?*
>
> **Authors response**:
>
> Yes, in a successful experiment, a part of a molecule is attached through non-specific adsorption to the tip of the AFM cantilever, while another part is non-specifically adsorbed onto the substrate. It is possible that multiple proteins are attached to the cantilever and thus leading to data that resulted from multiple proteins being pulled at the same time.
>
> *Q2. As far as I understand, we pull the proteins and observe their unfolding - how are the proteins attached to the head with which we pull?*
>
> **Authors response**:
>
> A part of the protein molecule is attached to the tip of the AFM cantilever when it is brought close to the molecule on the substrate and an indentation force of 1000-2000pN is applied to it. More experimental details can be found in Figure 1 on Page 3 of our paper.
>
> *Q3. How do we make sure that the molecules are always attached to the pulling head at the same residue?*
>
> **Authors response**:
>
> One portion of the protein randomly attaches to the substrate and the rest of the molecule is left free to interact with the cantilever tip (see Figure 1 on Page 3 of our paper). It is possible for different section of the protein molecule to attach to the cantilever tip (Figure 1b on Page 3), adding further complexity to the classification task.
>
>
> *Q4. Does the medium in which we measure the pulling force affect the force measurement outcomes?*
>
> **Authors response**:
>
> Yes, the medium can impact the bio-chemistry and bio-physics of the protein. We ensure consistency by using the same medium across all measurements. The experimental data we use has been collected by biologists [1, 2, 3], and we carefully verify that the chosen medium is both relevant and validated by experts in the field.
>
> [1] Rajaganapathy, Sivaraman, et al. "Distinct mechanical properties in homologous spectrin-like repeats of utrophin." Scientific reports 9.1 (2019): 5210.
>
> [2] Ramirez, Maria Paz, et al. "Phosphorylation alters the mechanical stiffness of a model fragment of the dystrophin homologue utrophin." Journal of Biological Chemistry 299.2 (2023).
>
> [3] Cailong Hua, Rebecca A. Slick, Joseph Vavra, Joseph M. Muretta, James M. Ervasti, and Murti V. Salapaka. Two operational modes of atomic force microscopy reveal similar mechanical properties for homologous regions of dystrophin and utrophin, May 2024. URL https://www.biorxiv. org/content/10.1101/2024.05.18.593686v1. Pages: 2024.05.18.593686 Section: New Results.
>
> *Q5. Why would we have the same protein repeated multiple times in a chain? Are they attached together in some fashion?*
>
> **Authors response**:
>
> Each protein has multiple domains and can be considered a chain with multiple domains. For example, structurally, dystrophin is composed of four major domains: an amino terminal (NT) actin-binding domain (ABD1), a large central rod domain with 24 triple helical spectrin-like repeats (SLRs) interspersed with 4 hinge domains, including a second actin-binding domain (ABD2), a cysteine-rich domain binding with the transmembrane dystroglycan complex, and a carboxy-terminal (CT) domain (see Figure 1 of [1]). Except for the engineered protein titin, the structures of the proteins shown in our study are as-is and are not engineered by the authors.
>
> [1] Ramirez, Maria Paz, et al. "Phosphorylation alters the mechanical stiffness of a model fragment of the dystrophin homologue utrophin." Journal of Biological Chemistry 299.2 (2023).

---

> > ### Comment · Reviewer_Mv6G · 2024-11-25
> > **Response by Reviewer**
> >
> > Thank you for the detailed answers to my many questions - this is very clarifying.
> >
> > It seems that I had 3 fundamental misunderstandings (e.g., that the measured proteins would always be repeats) and their clarification change my assessment.
> >
> > I change my assessment of "no novelty in application" to "small novelty in application," given that prior work applied ML classifiers to single molecule force spectroscopy but not to specific pulling measurements.
> >
> > Better understanding the tasks lets me see that it is not as trivially solved as I assumed. The technical novelty of the approach is limited, but it is an effective solution for an important task. I think many conference attendees would find value in reading this paper or attending the poster - I thus change my recommendation to acceptance.

---

> > > ### Author Response · Authors · 2024-11-26
> > >
> > > Thank you for your careful review of our response and for raising our score.

---

### Note · Authors · 2025-01-22

I have read and agree with the venue's withdrawal policy on behalf of myself and my co-authors.